# Plakoglobin is a mechanoresponsive regulator of naive pluripotency

Timo N. Kohler [1,2,12], Joachim De Jonghe[1,12], Anna L. Ellermann [1,12], Ayaka Yanagida[2,3,4], Michael Herger[1], Erin M. Slatery [5,6], Antonia Weberling [1,5], Clara Munger [1,5,6], Katrin Fischer [1], Carla Mulas[2,7,11], Alex Winkel[5], Connor Ross[2,5,8], Sophie Bergmann[5,6], Kristian Franze[5,9,10], Kevin Chalut [2,11], Jennifer Nichols[2,5,8], Thorsten E. Boroviak [2,5,6] ✉ & Florian Hollfelder [1] ✉

Biomechanical cues are instrumental in guiding embryonic development and cell differentiation. Understanding how these physical stimuli translate into transcriptional programs will provide insight into mechanisms underlying mammalian pre-implantation development. Here, we explore this type of regulation by exerting microenvironmental control over mouse embryonic stem cells. Microfluidic encapsulation of mouse embryonic stem cells in agarose microgels stabilizes the naive pluripotency network and specifically induces expression of Plakoglobin (*Jup*), a vertebrate homolog of β-catenin. Overexpression of Plakoglobin is sufficient to fully re-establish the naive pluripotency gene regulatory network under metastable pluripotency conditions, as confirmed by single-cell transcriptome profiling. Finally, we find that, in the epiblast, Plakoglobin was exclusively expressed at the blastocyst stage in human and mouse embryos – further strengthening the link between Plakoglobin and naive pluripotency in vivo. Our work reveals Plakoglobin as a mechanosensitive regulator of naive pluripotency and provides a paradigm to interrogate the effects of volumetric confinement on cell-fate transitions.

Cells integrate chemical and physical stimuli from the extracellular environment to regulate cell fate and function. Even the physical properties of a cell itself, including cortical stiffness[1], membrane tension[2,3], and intracellular crowding[4], are subject to dynamic changes in response to external cues, thereby integrating microenvironmental information into a physiological response[5,6]. This intricate pattern of regulation is of particular importance during embryogenesis, when pluripotent cells are faced with the complex task of undergoing spatiotemporal lineage specification to generate an entire embryo.

The initial state of this process, naive pluripotency, emerges in the epiblast of the mouse blastocyst prior to implantation and can be captured in vitro as mouse embryonic stem cells (mESCs)[7,8]. mESCs

[1]Department of Biochemistry, University of Cambridge, Hopkins Building, Tennis Court Road, Cambridge CB2 1QW, UK. [2]Wellcome Trust – Medical Research Council Stem Cell Institute, University of Cambridge, Jeffrey Cheah Biomedical Centre, Puddicombe Way, Cambridge CB2 0AW, UK. [3]Department of Veterinary Anatomy, Graduate School of Agriculture and Life Sciences, The University of Tokyo, Tokyo 113-8657, Japan. [4]Stem Cell Therapy Laboratory, Advanced Research Institute, Tokyo Medical and Dental University, 1-5-45 Yushima, Bunkyo-ku, Tokyo 113-8510, Japan. [5]Department of Physiology, Development and Neuroscience, University of Cambridge, Cambridge CB2 3DY, UK. [6]Centre for Trophoblast Research, University of Cambridge, Cambridge CB2 3EG, UK. [7]Randall Centre for Cell and Molecular Biophysics, King's College London, London SE1 1UL, UK. [8]MRC Human Genetics Unit, Institute of Genetics and Cancer, The University of Edinburgh, Crewe Road, Edinburgh EH4 2XU, UK. [9]Institute of Medical Physics, Friedrich-Alexander-Universität Erlangen-Nürnberg, Henkestr. 91, 91052 Erlangen, Germany. [10]Max-Planck-Zentrum für Physik und Medizin, 91054 Erlangen, Germany. [11]Present address: Altos Labs, Cambridge Institute of Science, Cambridge, UK. [12]These authors contributed equally: Timo N. Kohler, Joachim De Jonghe, Anna L. Ellermann. ✉e-mail: teb45@cam.ac.uk; fh111@cam.ac.uk

correspond to the pre-implantation epiblast[9] and retain full developmental capacity[10–12] when injected into a host embryo[11,13]. This naive 'ground state' of pluripotency can be sustained indefinitely in a defined culture regime termed 2i/LIF, where self-renewal is mediated through inhibition of mitogen-activated kinase kinase (MEK; by PD0325901, PD) and glycogen synthase kinase-3 (GSK-3; by CHIR99021, CH), as well as stimulation of JAK/STAT signaling by leukemia inhibitory factor (LIF)[12]. Inhibition of the MEK/ERK cascade suppresses developmental progression toward post-implantation stages[14,15]. GSK-3 inhibition stabilizes β-catenin, which supports the naive pluripotency circuitry through de-repression of TCF7L1 (also known as TCF3)[16,17] and LIF stimulates transcription of key pluripotency factors *Klf4* and *Tfcp2l1*[18,19]. Any two of the three 2i/LIF components (PD, CH, and LIF) are sufficient to maintain naive pluripotency[20]. In addition to chemical and cytokine-mediated signals, biomechanical cues are emerging as key players in the regulation of embryonic[21–25] and adult stem cells[4,26–28]. However, the identity of mechanoresponsive genes in mESCs and the mechanisms transforming physical responses into transcriptional programs remain poorly understood.

mESCs become responsive toward mechanical stress[29] and decrease membrane tension upon exit from naive pluripotency[3]. Membrane tension regulates the endocytic uptake of MEK/ERK signaling components and thus somatic cell-fate acquisition[2]. Recently, volumetric compression has been identified as a physical cue to promote self-renewal of intestinal stem cells in organoid cultures[4], and dedifferentiation of adipocytes[30], via WNT/β-catenin signaling. Volumetric compression induces molecular crowding, which stabilizes LRP6 signalosome formation and consequently elevates WNT/β-catenin signaling[4]. In the blastocyst, hydrostatic pressure also plays a critical role in embryo size regulation and cell fate specification[31,32]. However, little is known about the effects of volumetric compression and spatial confinement on naive pluripotent mESCs.

Considering the powerful effect of WNT/β-catenin signaling on naive pluripotency[16,33–35] and its link to molecular crowding, we sought to systematically interrogate volumetric confinement in mESCs. Biomimetic culture systems that emulate volumetric confinement found in the developing blastocyst may provide a route to analyze developmental processes under conditions of a crowded native environment in the embryonic niche. Here, we generated spherical agarose microgels[36–39] to encapsulate naive pluripotent mESCs and analyze their transcriptional and morphological development. We identify Plakoglobin (*Jup*)[40], a vertebrate homolog of β-catenin, as a mechanoresponsive gene and potent regulator of naive pluripotency. Plakoglobin supports pluripotency independently of β-catenin. Finally, we found Plakoglobin's association with naive pluripotency extends beyond the mouse model and was also evident in human naive pluripotent stem cells and the naive pre-implantation epiblast of human embryos.

## Results

### Microgel culture of embryonic stem cells stabilizes the pluripotent state

To investigate the effects of 3D-culture on pluripotency we encapsulated mouse ESCs into agarose microgels (Fig. 1a, b). This was performed at 37 °C in microfluidic flow-focusing chips by generating monodisperse water-in-oil droplets containing liquid low-melting agarose that, upon cooling on ice, formed biologically inert, soft, and spherical 3D hydrogel scaffolds (Figs. 1a, S1a, b). The resulting cell-containing microgels were cultured under self-renewing (naive pluripotent: 2i/LIF; metastable pluripotent: serum/LIF) or differentiating (N2B27) conditions as suspension cultures (Figs. 1c, S1c, d). Encapsulated mESCs in 2i/LIF displayed homogeneous expression of the general pluripotency marker OCT4 (gene name *Pou5f1*) and the naive pluripotency marker KLF4. In N2B27, these were downregulated and absent respectively, while the differentiation marker OTX2 was upregulated when cultured for 48 h (Figs. 1c, S1e). Interestingly, when

metastable pluripotent (serum/LIF) mESCs were cultured in microgels, the cells formed tightly packed colonies with indistinguishable cell boundaries and acquired a spherical morphology, similar to that seen in dome-shaped naive cells (2i/LIF) and unlike the cells cultured on plastic in serum/LIF (Figs. 1b, S1c, d). To determine the effects of 3D microgel culture on pluripotent mESCs, we utilized the *Rex1::GFPd2* reporter cell line (RGd2) expressing a destabilized green fluorescent protein (GFP) from the endogenous REX1 (gene name *Zfp42*) locus as a quantitative real-time readout for naive pluripotency[16] (Fig. 1d). In this system, loss of Rex1-GFPd2 indicates exit from naive pluripotency[16,33] (Figs. 1d, S1f, g). Consistent with previous reports[41], 2i/LIF induced uniformly high (~99% of cells) reporter expression in 2D culture, in contrast to serum/LIF which had a bimodal distribution (Fig. 1e, f). Encapsulated mESCs in 2i/LIF showed homogenous (~99% of cells) Rex1-GFPd2 levels, similar to conventional mESCs cultured in 2i/LIF in 2D. However, microgel suspension culture in serum/LIF revealed a ~20% increase in Rex1-GFPd2 expression compared to serum/LIF on gelatin-coated tissue culture plastic (Fig. 1f). When assessing the protein levels of other naive pluripotency markers, we found that these behaved similarly: for example, KLF4 protein levels increased in serum/LIF microgel-cultured mESCs compared to tissue culture plastic (Fig. 1g). Next, we tested the developmental potential of encapsulated mESCs via chimeric blastocyst integration. H2B-tq labeled mESCs were cultured for 48 h in agarose microgel suspension culture, released from the gels by agarase treatment and micro-injected (3–5 cells/embryo) into 8-cell stage host embryos (Fig. 1h). Microgel-cultured mESCs robustly colonized the pluripotent epiblast compartment, as indicated by immunofluorescence staining of the colocalized epiblast marker SOX2, with an efficiency of more than 70% (Fig. 1h). We conclude that 3D agarose microgel culture is suitable for the culture of naive pluripotent (2i/LIF) mESCs and stabilizes pluripotency under metastable pluripotent (serum/LIF) culture conditions.

### Microgel suspension culture induces Plakoglobin expression

To delineate the microgel-induced changes in the transcriptional program of pluripotent cells, we performed bulk RNA-seq of naive mESCs cultured on tissue culture plastic *versus* microgel encapsulated mESCs in naive pluripotency-supporting 2i/LIF. Pearson correlation and principal component analysis (PCA) confirmed that samples separated according to experimental conditions (Figure S2a, b). General pluripotency factors remained robustly expressed in both conditions, with the core pluripotency factors *Pou5f1* and *Sox2* being elevated in 3D-microgel cultured cells (Fig. 2a). In addition, we observed a global increase in expression of naive pluripotency factors such as *Esrrb*, *Klf2*, *Klf4*, *Klf5*, *Tbx3*, and *Tfcp2l1* in encapsulated mESCs (Fig. 2b). A transcriptome-wide comparison showed that some of the most differentially expressed genes were the naive pluripotency factors *Klf2*, *Klf4*, and components of the WNT/β-catenin signaling pathway, including *Sfrp1* and *Esrrb*[42] (naive pluripotency-associated transcription factor and downstream target of β-catenin) as well as the β-catenin vertebrate homolog *Jup* (Plakoglobin) (Fig. 2c). We performed gene set enrichment analysis (GSEA) of 2D-tissue culture plastic *versus* encapsulated mESCs and obtained enrichments for "PluriNetWork", "Regulation of Actin Cytoskeleton" and "WNT Signaling" for mESCs cultured in agarose microgels (Figure S2c, d). Considering the important role of the WNT/β-catenin signaling for self-renewal and pluripotency in the mouse[16] and β-catenin's crucial role in the formation of adherens junctions, we examined individual members of adherens junction-associated genes: *Cdh1* (E-cadherin), *Ctnna1* (α-catenin), *Ctnnb1* (β-catenin), *Jup* (Plakoglobin/γ-catenin), and *Ctnnd1* (p120/δ-catenin). *Cdh1*, *Ctnna1*, and *Ctnnb1* expression remained mostly unchanged, while *Jup* and *Ctnnd1* were significantly upregulated (Fig. 2d). *Jup* showed the most prominent (~6-fold) increase in microgels. Immunofluorescence staining confirmed significant

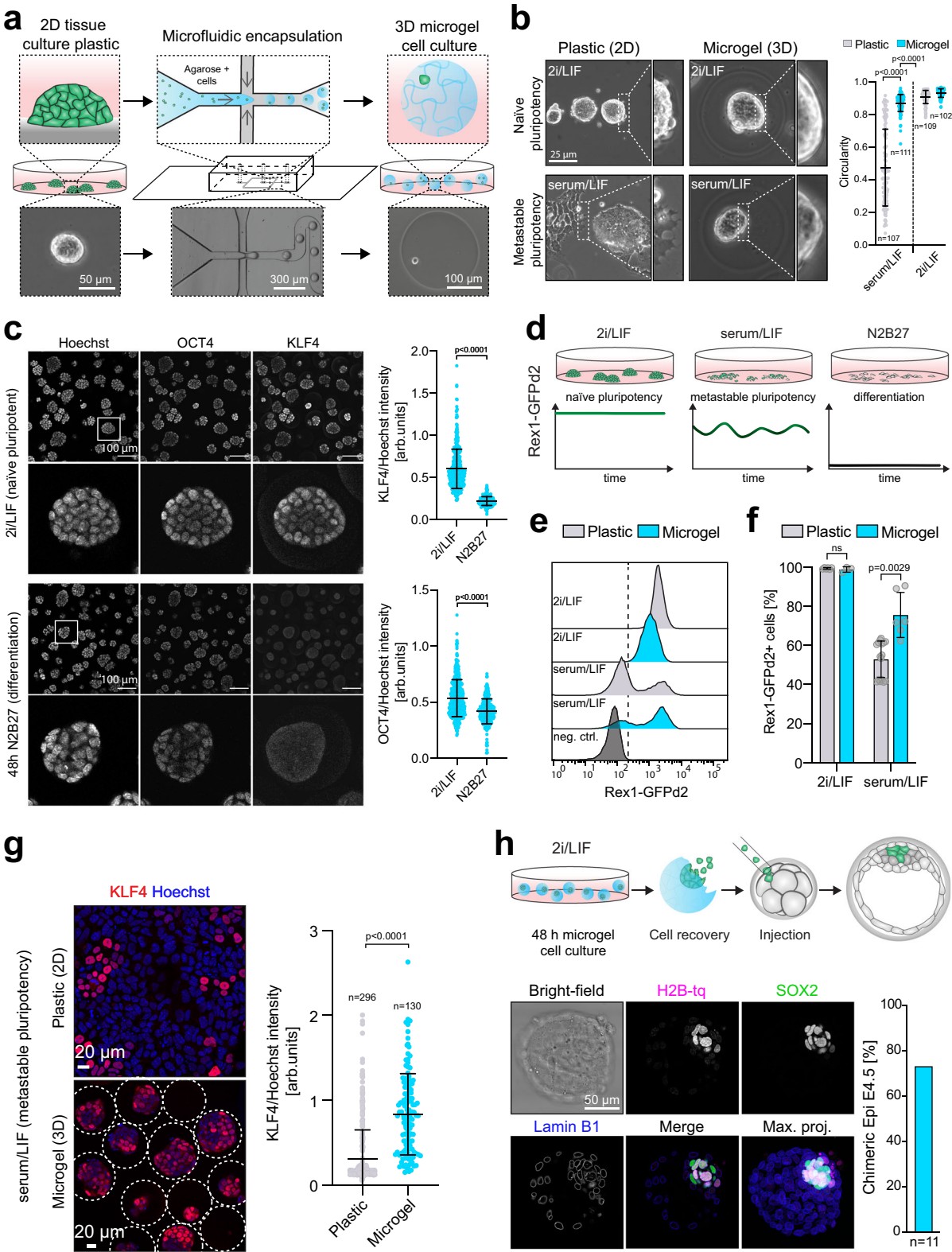

upregulation of Plakoglobin (*Jup*) upon agarose microgel culture also at the protein level and demonstrated localization to the cell cortex (Figs. 2e, S2e, f). β-catenin was expressed in mESCs cultured in both conditions, although we noted reduced cortical protein levels in microgel cultured cells (Figs. 2f, S2f). These results demonstrate that microgel culture induces Plakoglobin, which localizes to the cellular cortex at the expense of β-catenin.

**Plakoglobin expression is regulated via microgel-mediated volumetric confinement**

To delineate microgel-induced volumetric confinement from general effects of 3D-culture, we compared mESCs cultured under naive conditions in microgel with ESCs cultured in hanging drops (Fig. 3a, b). Hanging drop culture allows mESCs to grow in suspension as 3D-aggregates, but in the absence of confinement from the microgel

**Fig. 1 | Agarose microgel encapsulation of mouse ESCs supports naive pluripotency. a** Schematic illustration of the microfluidic cell encapsulation process. Tissue culture plastic (2D) mouse ESCs were single cell separated and microfluidically compartmentalized into agarose-in-oil microdroplets. Cell-laden agarose microgels (3D) were cultured in a static suspension culture in low adhesion tissue culture plates. **b** Phase contrast images of mESCs cultured on tissue culture plastic or encapsulated in microgels in naive pluripotent (2i/LIF) or metastable pluripotent (serum/LIF) conditions. Circularity of mESC colonies was measured using Fiji[107]. The *p*-values were calculated using a two-tailed unpaired *t*-test with Welch's correction; 3 independent experiments. Error bars indicate the mean and standard deviations. **c** Confocal immunofluorescence images of microgel-encapsulated mESCs stained for the general pluripotency marker OCT4 and the naive pluripotency marker KLF4 after being cultured for 48 h in 2i/LIF (naive pluripotent) or N2B27 (differentiation). KLF4 and OCT4 intensities (as shown on the right) were normalized to Hoechst. Error bars indicate the mean and standard deviations. *n* = 400 (2i/LIF), *n* = 285 (N2B27). The *p*-values were calculated using a two-tailed unpaired *t*-test with Welch's correction; 3 independent experiments. **d** The *Rex1::GFPd2* (RGd2) reporter

system allows near real-time analysis of pluripotency (destabilized GFP with a half-life of 2 h). Homogenous expression (naive in 2i/LIF), heterogeneous expression (metastable in serum/LIF), and loss of expression upon exit from pluripotency (N2B27). **e, f** Flow cytometric analysis of RGd2 mESCs cultured on plastic compared to microgels. Negative control: wild-type mESCs. Error bars indicate the mean and standard deviations. The *p*-values were calculated using a two-tailed unpaired *t*-test with Welch's correction. *N* = 6 (plastic 2i/LIF), *N* = 3 (microgel 2i/LIF), *N* = 16 (plastic serum/LIF), *N* = 6 (microgel serum/LIF). **g** Confocal immunofluorescence images of serum/LIF cultured (microgel and plastic) mESCs stained for KLF4. KLF4 intensities (as shown on the right) were normalized to Hoechst. Error bars indicate the mean and standard deviations. The *p*-values were calculated using a two-tailed unpaired *t*-test with Welch's correction. *n* = 296 (plastic), *n* = 130 (microgel), 3 independent experiments. **h** Injection of microgel-cultured naive mESCs (nuclear H2B-tq reporter) into 8-cell stage embryos led to ~73% (*N* = 8/11) blastocysts chimeras 48 h after in vitro culture. Injected cells were identified via the H2B-tq reporter. Cells were stained for Lamin B1 (outlines all nuclei) and SOX2 (epiblast).

scaffold (Fig. 3a). Remarkably, microgel encapsulated cells, and to a much less pronounced extent cells cultured in hanging drops, showed an upregulation of Plakoglobin−an effect observed as early as 48 h after encapsulation (Figs. 3b, S3a). In both conditions the naive marker KLF4 remained strongly expressed (Fig. 3b). We then investigated Plakoglobin and β-catenin expression of mESCs grown under naive pluripotent conditions in 2i/LIF on tissue culture plastic, in microgels and in suspension culture (similar to hanging drops but allowing higher throughput analysis) (Fig. 3a, c–e). We found that Plakoglobin and β-catenin were upregulated when maintained as a suspension culture compared to 2D plastic. In contrast, when encapsulated into microgels an inverse relationship was observed: Plakoglobin levels were significantly increased compared to plastic or suspension culture, whereas β-catenin levels were reduced (Fig. 3c, e). To address the role played by confinement in the regulation of Plakoglobin, we propagated cells on tissue culture plastic and used a bulk agarose gel (Plastic + Gel) to induce confining conditions akin to the microgels (Fig. 3a, f). Consistent with the observations seen in microgel culture, mESCs confined by a bulk agarose gel also showed upregulation of Plakoglobin (Fig. 3f). Interestingly, when cultured on commercially available 2D PDMS gels with varying stiffness (1.5 kPa, 15.0 kPa, and 28.0 kPa) but without additional confinement in 3D, Plakoglobin levels remained low and were mostly unaffected by matrix stiffness (Figure S3b). These results suggest that Plakoglobin upregulation and maintained expression are dependent on gel-mediated volumetric confinement, and to a lesser degree by 3D culture itself.

## Plakoglobin expression is a defining feature of the naive pre-implantation epiblast in mouse and human embryos

Considering that volumetric confinement induced Plakoglobin expression in microgel-encapsulated pluripotent mESCs, we sought to examine *Jup*/Plakoglobin expression and localization in the pre- and peri-implantation mouse embryo during the establishment of naive pluripotency in vivo. Meta-analysis of RNA-seq data[10,43] showed that *Jup* transcription was highest in the pluripotent compartment (E3.5-ICM and E4.5-EPI) of the pre-implantation mouse embryo (Figure S4a), consistent with *Jup* expression in the naive pluripotent mESCs. Notably, *Jup* transcription was sustained in diapause, a facultative developmental arrest at the blastocyst stage[44,45], but downregulated in the post-implantation epiblast (Figure S4a). This stands in contrast to *Ctnnb1*, which was robustly expressed throughout pre-implantation development and further upregulated upon implantation (Figure S4b). At the protein level, we did not detect Plakoglobin at the 8-cell stage (E2.5) but obtained a robust cortical signal in both the trophectoderm and the ICM at the blastocyst stage (E3.5-E4.5), during which pluripotency is established (Fig. 4a). Strikingly, Plakoglobin was absent in the primed pluripotent, epithelialized epiblast of the early post-

implantation egg-cylinder at (E5.5-E6.5), while it remained in the extraembryonic ectoderm (ExE) (Fig. 4b).

In contrast, β-catenin did not exhibit stage-specific patterns and was localized cortically throughout all stages analyzed (Fig. 4c, d), in agreement with the transcriptome data (Figure S4b). We confirmed these findings in vitro by culturing mouse epiblast-derived stem cells (EpiSCs) which correspond to the post-implantation epiblast[46,47]. EpiSCs did not express Plakoglobin, but were positive for β-catenin and E-cadherin (Figure S4c), thus phenocopying the expression dynamics in the embryo. Transcriptionally, EpiSCs have been shown by meta-analysis[48] to exhibit variable and overall reduced *Cdh1* and *Jup* expression compared to naive pluripotent mESCs (Figure S4d). *Ctnnb1* expression remained constant whereas the naive marker *Zfp42* was not expressed in EpiSCs. To investigate whether Plakoglobin expression was restricted to the pluripotent epiblast cells of the ICM, we performed immunofluorescence staining in hatched blastocysts at E4.25 for Plakoglobin, the epiblast marker OCT4 and the primitive endoderm (PrE) marker SOX17 (Figs. 4e, S4e). Plakoglobin expression was detectable in SOX17-positive cells where the PrE had not sorted to the ICM surface and not yet formed a continuous epithelium. However, in slightly later stages when the PrE had proceeded to form a mature epithelium Plakoglobin expression was downregulated (Figs. 4e, f, S4e). Due to the high degree of evolutionary conservation of β-catenin and Plakoglobin[49–51], we investigated Plakoglobin's expression pattern in human embryos. Immunofluorescence staining of Plakoglobin revealed its absence in early cleavage embryos at the 4-cell stage, when β-catenin was clearly present (Fig. 4g). However, consistent with mouse development, human embryos at the blastocyst stage, robustly displayed cortical Plakoglobin signal in the trophectoderm and the pluripotent epiblast (Figs. 4h, S4f). Co-staining of Plakoglobin with the epiblast markers SOX2 and the hypoblast marker SOX17 showed that hypoblast fate is linked to a reduction in Plakoglobin levels (Fig. 4h). In the mouse epiblast and in pluripotent stem cells Plakoglobin is downregulated during the naive (pre-implantation) to primed (post-implantation) transition. However, early human post-implantation samples are mostly inaccessible. Therefore, we leveraged marmoset embryos and human pluripotent stem cells (hPSCs) as model systems to investigate Plakoglobin during primate post-implantation development (Figs. 4i, j, S4g). In the post-implantation Carnegie stage 6 (CS6) marmoset embryo, the pluripotent embryonic disc is demarcated by SOX2 and OTX2 expression and was negative for Plakoglobin (Fig. 4i, S4g). We then analyzed *JUP* and *CTNNB1* expression in a marmoset post-implantation spatial transcriptome dataset[52], further validating the downregulation of *JUP* upon implantation (Figure S4h) and consistent with immunofluorescence and transcriptional data from mouse embryo development (Figs. 4a−d, S4a, b). In vitro, the embryo-derived naive hPSC line HNESI[53] expressed OCT4, the naive pluripotency

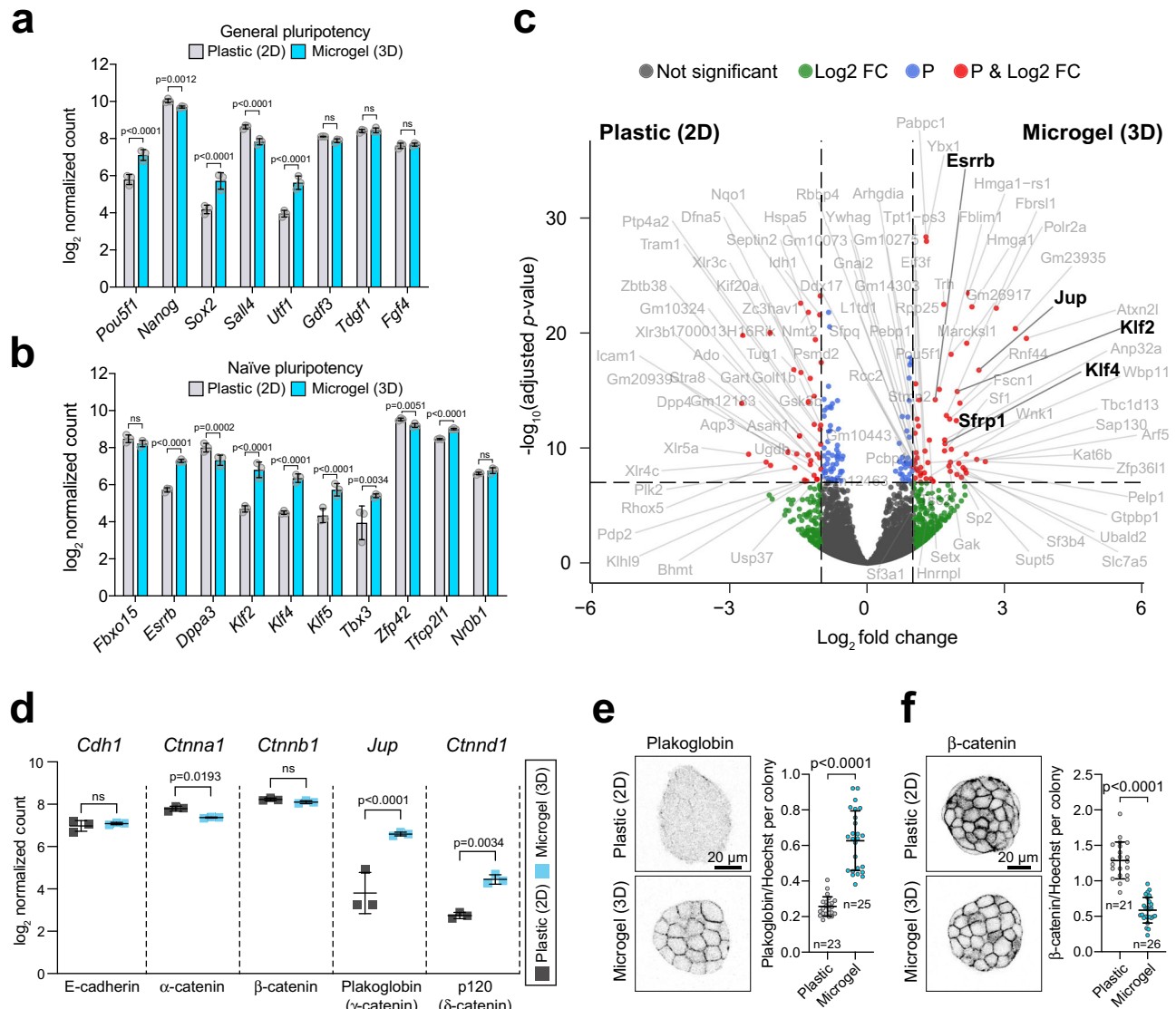

**Fig. 2 | Transcriptomic analysis of microgel-encapsulated mESCs reveals upregulation of Plakoglobin. a, b** Bulk RNA-seq analysis of general (**a**) and naive (**b**) pluripotency-associated genes for mESCs cultured on plastic (2D, gray) or encapsulated in microgels (3D, blue) under 2i/LIF culture conditions. Error bars indicate the mean and standard deviations. 3 independent experiments; statistical significance levels are Bonferroni-adjusted $p$-values computed via two-tailed differential gene expression analysis using the DESeq2 tool. **c** Volcano plot representing the list of differentially expressed genes between mESCs cultured on tissue culture plastic (negative $\log_2$ fold changes in gene expression) or in agarose microgels (positive $\log_2$ fold changes in gene expression) in 2i/LIF. $\log_2$ fold change cut-off was |$\log_2$ fold change| > 1 and Bonferroni-adjusted $p$-values cut-off was $10^{-7}$. NS stands for not-significant (gray), $\log_2$ FC (green color) indicates gene that passed the $\log_2$ fold change cut-off, P (blue) indicates genes that passed $p$-value cut-off, P&$\log_2$ FC (red) indicates genes that passed $\log_2$ FC and $p$-value cut-offs (see full list of differentially expressed genes in Supplementary Data 1). **d** 3D microgel culture led to a significant upregulation at gene expression level of Plakoglobin

(*Jup*) and p120 (*Ctnnd1*) compared to 2D plastic culture conditions. Error bars indicate the mean and standard deviations. 3 independent experiments; statistical significance levels are Bonferroni-adjusted $p$-values computed via two-tailed differential gene expression analysis using the DESeq2 tool. **e** Confocal immunofluorescence images of naive pluripotent (2i/LIF) mESCs cultured on plastic and encapsulated in microgels stained for Plakoglobin. Plakoglobin levels were quantified for colonies grown on plastic ($n = 23$) and in microgels ($n = 25$). Error bars indicate the mean and standard deviations. The $p$-values were calculated using a two-tailed unpaired $t$-test with Welch's correction; 3 independent experiments. **f** Confocal immunofluorescence images of naive pluripotent (2i/LIF) mESCs culture on plastic and encapsulated in microgels stained for β-catenin. β-catenin levels were quantified for colonies grown on plastic ($n = 21$) and in microgels ($n = 26$). Error bars indicate the mean and standard deviations. The $p$-values were calculated using a two-tailed unpaired $t$-test with Welch's correction; 3 independent experiments.

marker KLF17 with high levels of Plakoglobin (Fig. 4j, S4i). Upon capacitation[54], HNES1 cells downregulated Plakoglobin and were KLF17 negative whilst OCT4 expression was maintained (Figs. 4j, S4i). In contrast, no Plakoglobin was detected in the primed hPSC line H9. The combination of cross-species transcriptional analysis and immunofluorescence data, allows the conclusion that Plakoglobin is an evolutionarily conserved feature of the naive pre-implantation pluripotent epiblast.

## Plakoglobin overexpression promotes the naive pluripotency gene regulatory network

Considering that Plakoglobin was one of the most differentially expressed genes in microgel-cultured mESCs, we sought to test whether Plakoglobin overexpression could mimic the pluripotency-promoting effects of the microgel suspension culture system. We generated Plakoglobin-overexpressing (PG OE) cells by genomic integration of a *Jup-2A-mCherry* cassette under the control of a constitutive

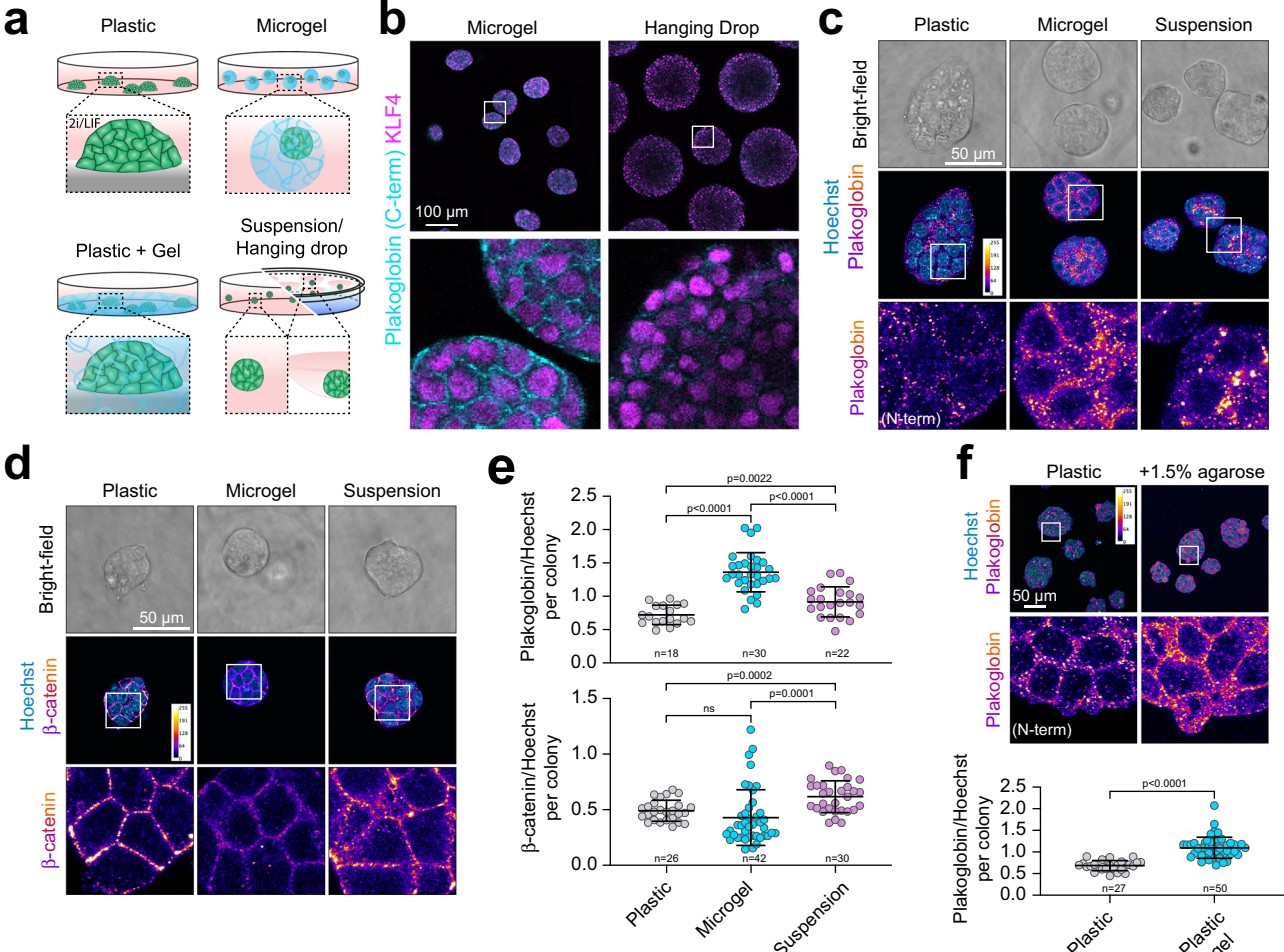

**Fig. 3 | Microgel-mediated volumetric confinement induces mechanoresponsive Plakoglobin expression in mESCs. a** Schematic cell culture formats: conventional 2D-tissue culture plastic, agarose microgel encapsulation, 2D-tissue culture plastic overlaid with a bulk agarose gel and suspension/hanging drop culture. **b** Confocal immunofluorescence images of naive pluripotent (2i/LIF) mESCs cultured as hanging drops or microgel-encapsulated. Cells were stained for Plakoglobin (C-term) and KLF4 (naive pluripotency marker) after 48 h in culture. Representative of 3 independent experiments. **c, d** Confocal immunofluorescence images of naive pluripotent (2i/LIF) mESCs on tissue culture plastic (2D), in microgels (3D confined) or as suspension culture (3D unconfined) after 48 h. Cells were stained for (**c**) Plakoglobin and (**d**) β-catenin. **e** Quantification of Plakoglobin and β-catenin intensities per colony normalized by Hoechst. Error bars indicate the mean and standard deviations. *p*-values were determined by a two-tailed unpaired *t*-test with Welch's correction; 3 independent experiments. **f** mESCs were cultured for 48 h on tissue culture plastic or on plastic overlaid with 1.5% agarose. Immunofluorescence analysis of Plakoglobin (N-term) showed an increase upon gel-mediated confinement in 2D. Error bars indicate the mean and standard deviations. *n* = 27 (Plastic), *n* = 50 (Plastic + 1.5% agarose). The *p*-values were calculated using a two-tailed unpaired *t*-test with Welch's correction; 3 independent experiments.

promoter in the RGd2 reporter line using the piggyBac™ system (Fig. 5a). The resulting cell lines exhibited exogenous Plakoglobin expression via *Jup-2A-mCherry* and naive pluripotency via *Rex1::GFPd2*. We derived clonal mESC lines overexpressing Plakoglobin at high (PG$^{HIGH}$, clones #1-3) and low (PG$^{LOW}$, clones #1-3) levels (Figs. 5b, c, S5a). Consistent with previous reports[17,55], we found a reciprocal correlation between Plakoglobin and β-catenin levels by immunocytochemistry and Western blotting (Figs. 5d, e, S5g, h). Furthermore, PG$^{HIGH}$ cells showed higher nuclear Plakoglobin signal compared to PG$^{LOW}$ cells (Figs. 5d, S5b).

We examined the ability of Plakoglobin-overexpressing mESCs to sustain features of naive pluripotency by exposing the cells to metastable serum/LIF conditions. All PG$^{HIGH}$ clones cultured in serum/LIF readily sustained the tightly packed, dome-shaped morphology associated with naive pluripotency, as observed in 2i/LIF (Figs. 5c, S5a). PG$^{LOW}$ clones remained mostly as a monolayer (Fig. 5c), similar to control RGd2 cells (Figure S5e). Strikingly, flow cytometric analysis revealed that all PG$^{HIGH}$ clones displayed a uniformly positive Rex1-GFPd2 signal, indistinguishable from naive mESCs cultured in 2i/LIF

(Figs. 5f, S5a). In contrast, PG$^{LOW}$ cells maintained a bimodal GFP distribution, unless cultured in 2i/LIF (Figs. 5f, S5a). We also assessed the naive pluripotency markers KLF4 and TFCP2L1 (Figs. 5g, S5c), the core pluripotency factor OCT4 (Fig. 5h) and the mesoderm marker T (Brachyury) (Fig. 5h) in the presence or absence of LIF. Consistent with the homogeneous Rex1-GFPd2 signal, we found that 100% of PG$^{HIGH}$ cells in serum/LIF expressed the naive pluripotency factor KLF4 in comparison to ~22% of parental RGd2 cells (Fig. 5g). Upon LIF removal for 72 h, these values dropped to ~14 and 0%, respectively. The general pluripotency factor OCT4 was expressed in all conditions and slightly elevated in PG$^{HIGH}$ cells (Fig. 5h). The mesoderm marker T (Brachyury) was expressed in a subset (7% of cells) of serum/LIF cultured RGd2 cells and increased (39% of cells) after LIF removal (Fig. 5i). In contrast, only 3% of PG$^{HIGH}$ cells in serum/LIF expressed T (Brachyury), increasing to 7% after LIF removal. We repeated all experiments with polyclonal lines to exclude potential artifacts from clonal selection (Figure S5d–m). These results demonstrate that high Plakoglobin expression stabilizes the naive pluripotency network in metastable culture conditions

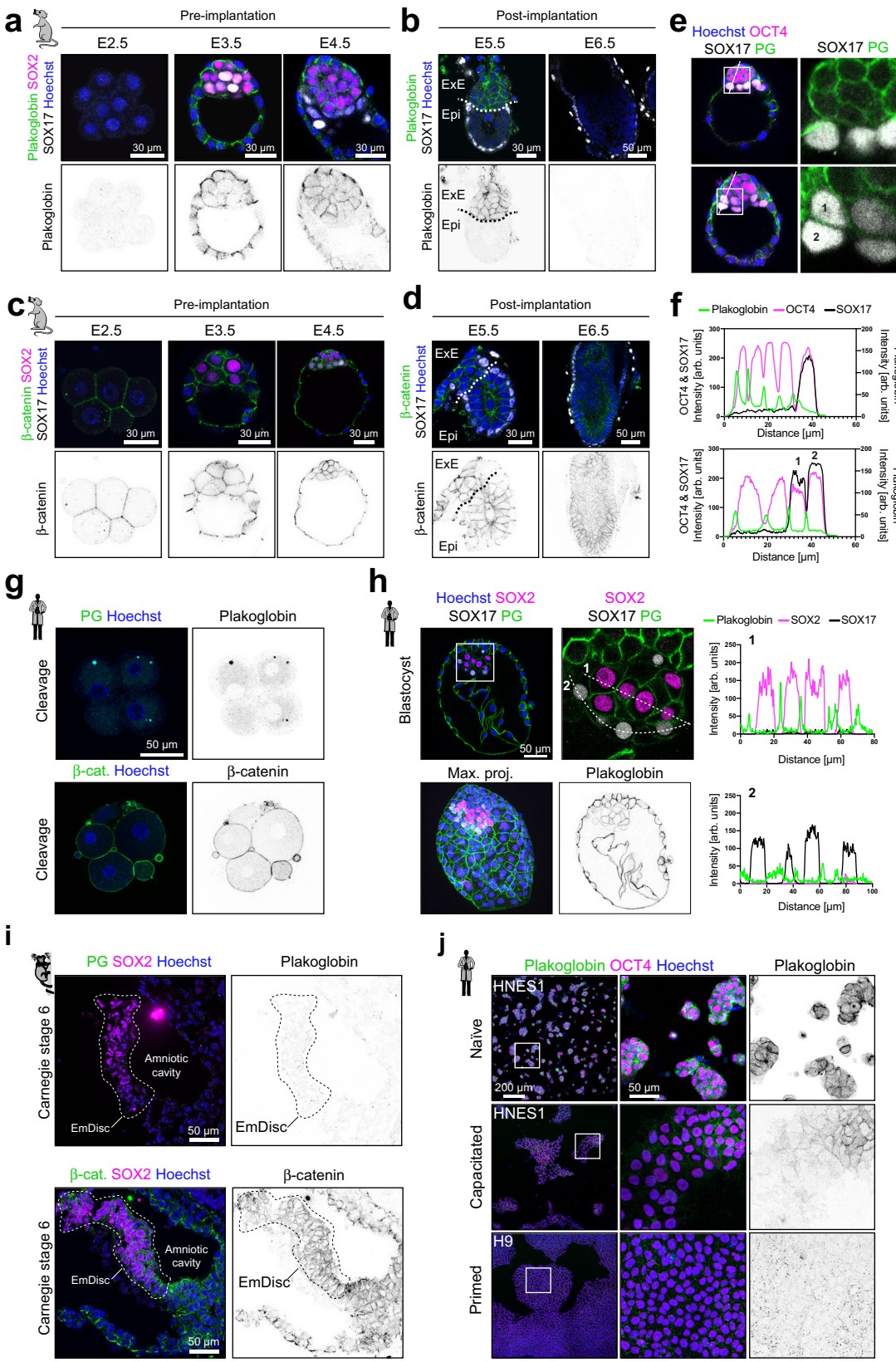

(serum/LIF), thereby recapitulating the effects of microgel suspension culture.

To comprehensively evaluate the effects of Plakoglobin overexpression on the pluripotency gene regulatory network, we performed single-cell RNA-seq using the inDrop workflow[56] for the following samples: RGd2 mESCs in 2i/LIF, RGd2 mESCs in serum/LIF,

encapsulated RGd2 mESCs in serum/LIF, clonal PG^LOW mESCs in serum/LIF and clonal PG^HIGH mESCs in serum/LIF (Fig. 6a). Pearson correlation coefficient computation between all samples revealed that PG^HIGH mESCs in serum/LIF were transcriptionally most correlation with the naive control cells in 2i/LIF (Fig. 6b). All remaining serum/LIF samples exhibited lower degrees of resemblance to naive control cells (RGd2

**Fig. 4 | Plakoglobin expression is evolutionarily conserved and associated with pre-implantation pluripotency. a** Confocal immunofluorescence images of pre-implantation embryos (E2.5, E3.5, and E4.5) stained for Plakoglobin, SOX2 (epiblast marker) and SOX17 (primitive endoderm marker, pre-implantation). Plakoglobin expression in the epiblast coincides with the emergence of naive pluripotency in the pre-implantation epiblast. Scale bar: 30 μm. Representative of $n \geq 3$ embryos. **b** Confocal immunofluorescence images of post-implantation embryos (E5.5 and E6.5) stained for Plakoglobin and SOX17 (visceral endoderm marker, post-implantation). In the post-implantation embryo Plakoglobin expression was restricted to the extraembryonic ectoderm (ExE) and completely absent in the epiblast (Epi) at E5.5. Scale bar: 30 μm. Representative of $n \geq 3$ embryos. **c** Confocal immunofluorescence images of pre-implantation embryos (E2.5, E3.5, and E4.5) stained for β-catenin, SOX2 (epiblast marker) and SOX17 (primitive endoderm marker, pre-implantation). β-catenin was detected in all cell lineages throughout pre-implantation development. Scale bar: 30 μm. Representative of $n \geq 3$ embryos. **d** Confocal immunofluorescence images of post-implantation embryos (E5.5 and E6.5) stained for β-catenin and SOX17 (visceral endoderm marker, post-implantation). β-catenin remained detectable in the post-implantation epiblast (Epi) and

extraembryonic ectoderm (ExE). Scale bar: 30 μm. Representative of $n \geq 3$ embryos. **e, f** Confocal immunofluorescence images of blastocysts (~E4.25) stained for Plakoglobin, OCT4 (epiblast marker), and SOX17 (primitive endoderm marker). Fluorophore intensities were quantified along the white line marked in the merged image and are shown in (**f**). Primitive endoderm cells lost Plakoglobin expression once sorted to the inner cell mass' surface. Scale bar: 30 μm. Representative of $n \geq 3$ embryos. **g** Representative confocal immunofluorescence images of human cleavage stage embryos stained for Plakoglobin ($N = 3$) and β-catenin ($N = 2$). Scale bar: 50 μm. **h** Confocal immunofluorescence images of a human blastocyst (day 7, $N = 4$) stained for SOX2 (epiblast marker), SOX17 (hypoblast marker) and Plakoglobin. Fluorophore intensities were quantified along the white line and are shown on the right. Scale bar: 50 μm. **i** Confocal immunofluorescence images of Carnegie stage 6 (CS6) post-implantation marmoset embryos stained for Plakoglobin (PG), β-catenin, and SOX2 (epiblast marker). Scale bar: 50 μm. **j** Confocal immunofluorescence images of human pluripotent stem cells. Naive HNES1 cells, capacitated HNES1 cells, and primed H9 cells were stained for Plakoglobin and OCT4. Representative of 3 independent experiments.

mESCs in 2i/LIF). Principal component analysis separated pluripotent cell populations (RGd2 mESCs in 2i/LIF, PG$^{HIGH}$ mESCs in serum/LIF, encapsulated RGd2 mESCs in serum/LIF) from differentiating cells (PG$^{LOW}$ mESCs in serum/LIF, RGd2, mESCs in serum/LIF) along the first principal component (Fig. 6c). We determined developmental trajectories by RNA latent time computation using dynamical RNA velocity as an input[57,58] (Fig. 6d–i), which revealed the presence of a continuum of cell states from naive to primed pluripotency (Fig. 6d–i). The core pluripotency factor *Pou5f1* (OCT4) was expressed in most cells, whereas expression of naive transcription factors including *Esrrb*, *Klf2*, *Tfcp2l1*, and *Zfp42* was restricted to PG$^{HIGH}$ cells and the 2i/LIF control (Fig. 6e, f). The early post-implantation markers *Pou3f1* (OCT6) and *Otx2*[10,59] and early differentiation markers such as *Anxa2*, *Krt18*, and *Tubb6*[60] were upregulated in RGd2 mESCs on plastic and PG$^{LOW}$ mESCs in serum/LIF conditions, but absent from RGd2 mESCs in agarose microgels (Fig. 6g). PG$^{HIGH}$ cells differed from all other serum/LIF cultured samples by their upregulation of naive pluripotency-associated threonine dehydrogenase (*Tdh*)[61,62] and downregulation of γ-actin (*Actg1*) (Figure S6b-d). 2i/LIF control cells separated from all serum/LIF samples (including PG$^{HIGH}$ cells) by their upregulation of *Vim*, *Scd2*, *Ldhb*, and downregulation of *Tmsb4x*, *Mycn*, and *Ldha* (Figure S6e, f). These data demonstrate that high Plakoglobin expression is more effective in sustaining naive pluripotency compared to microgel encapsulation alone, which mainly prevents the upregulation of early lineage specifiers.

To functionally assess the self-renewal and developmental potential of serum/LIF cultured PG$^{HIGH}$ mESCs, we compared RGd2 mESCs against PG$^{HIGH}$ mESCs using an in vitro clonogenicity and single-cell blastocyst chimera contribution assays. PG$^{HIGH}$ cells displayed high colony-forming efficiencies in 2i/LIF and serum/LIF, while the self-renewal capacity of serum/LIF cultured RGd2 mESCs was reduced upon replating into naive 2i/LIF culture conditions (Figure S6a). To examine the ability of PG$^{HIGH}$ mESCs to contribute to the blastocyst, we injected individual PG$^{HIGH}$ mESCs cultured in serum/LIF, control H2B-tq (serum/LIF) and naive H2B-tq (2i/LIF) mESCs into 8-cell stage embryos and cultured them in vitro until the blastocyst stage (Fig. 6j). Single-cell injections resulted in exclusive contribution to the epiblast and not the extraembryonic lineages (Fig. 6k). Remarkably, PG$^{HIGH}$ cells exhibited ~1.8-fold increased chimeric blastocyst contribution efficiency (~77%) compared to the wild-type cells in 2i/LIF (~41%) and serum/LIF (~40%) (Fig. 6l). Taken together, mESCs with high levels of exogenous Plakoglobin expression acquire a transcriptional state reminiscent of naive pluripotency that is retained even under serum/LIF conditions and cells in this state readily contribute to the developing blastocyst.

## Plakoglobin maintains naive pluripotency independently of β-catenin

To determine how Plakoglobin sustains naive pluripotency, we interrogated the signaling requirements of Plakoglobin-overexpressing mESCs for self-renewal (Fig. 7a). In 2i/LIF, the combination of any two of the three components (PD, CH, and LIF) is sufficient to maintain naive pluripotency[20]. Thus, we challenged self-renewal for more than five passages of wild type (RGd2), PG$^{LOW}$ and PG$^{HIGH}$ mESCs in medium supplemented with only one of the individual 2i/LIF components (Fig. 7b–d). Naive pluripotency was assessed through the Rex1-GFPd2 reporter by flow cytometry. In the presence of LIF-only, wild-type (RGd2) mESCs could not be passaged beyond day 6, consistent with previous observations[16]. PG$^{LOW}$ displayed reduced Rex1-GFPd2 levels, but could be propagated (Figs. 7b, S7b). Remarkably, PG$^{HIGH}$ mESCs cultured with LIF-only gave rise to dome-shaped morphology and displayed homogenous (>99%) Rex1-GFPd2 levels, indistinguishable from 2i/LIF cells (Figs. 7b, S7a). In PD-only conditions, wild-type (RGd2) and PG$^{LOW}$ mESCs were unable to self-renew and the cultures collapsed by day 4 (Figs. 7c, S7b). PG$^{HIGH}$ mESCs exhibited robust expression of Rex1-GFPd2 (~75–90%) and could be propagated throughout the time course (Figs. 7c, S7b). Wild type (RGd2) cultured in the presence of CH-only exhibited medium (>60%) Rex1-GFPd2 levels at day 6 and could not be stably propagated (Fig. 7d). PG$^{LOW}$ mESCs showed some heterogeneity but could be passaged until day 16 (Figs. 7d, S7b). Notably, PG$^{HIGH}$ mESCs failed to maintain pluripotency (Fig. 7d). Collectively, the single-component time course analysis demonstrated that Plakoglobin overexpression can sustain pluripotency in combination with PD or LIF, but not CH. To determine whether Plakoglobin can functionally replace β-catenin to maintain naive pluripotency, we cultured mESCs in the presence of the tankyrase inhibitor XAV939 (XAV) to abrogate nuclear β-catenin signaling[63]. We performed single-component (PD, CH, or LIF) time course analysis of PG$^{HIGH}$ and PG$^{LOW}$ mESCs in the presence of XAV. Self-renewal with LIF-only or PD-only in PG$^{HIGH}$ cells was unaffected by XAV treatment. Only PG$^{LOW}$ cells supplemented with CH, started to differentiate and lost GFP expression upon XAV treatment (Fig. 7e). In RGd2 cells microgel encapsulation could not compensate for the lack of exogenous Plakoglobin overexpression and cells rapidly downregulated the Rex1-GFP reporter upon culture with LIF, PD, or CH alone (Figure S7c). To confirm that Plakoglobin-mediated maintenance of pluripotency is truly independent of β-catenin, we generated clonal *Ctnnb1* (β-catenin) knockout PG$^{HIGH}$ cell lines: PG$^{HIGH}$ *Ctnnb1* KO#2, KO#10, and KO#12 (Figure S7d). By immunofluorescence analysis β-catenin was not detectable in clone #10 and #12 even though, for clone #2 a residual signal remained (Fig. 7f). Additional knockout validation by Western blotting confirmed the absence of β-catenin protein in clones #10 and #12 and indicates low

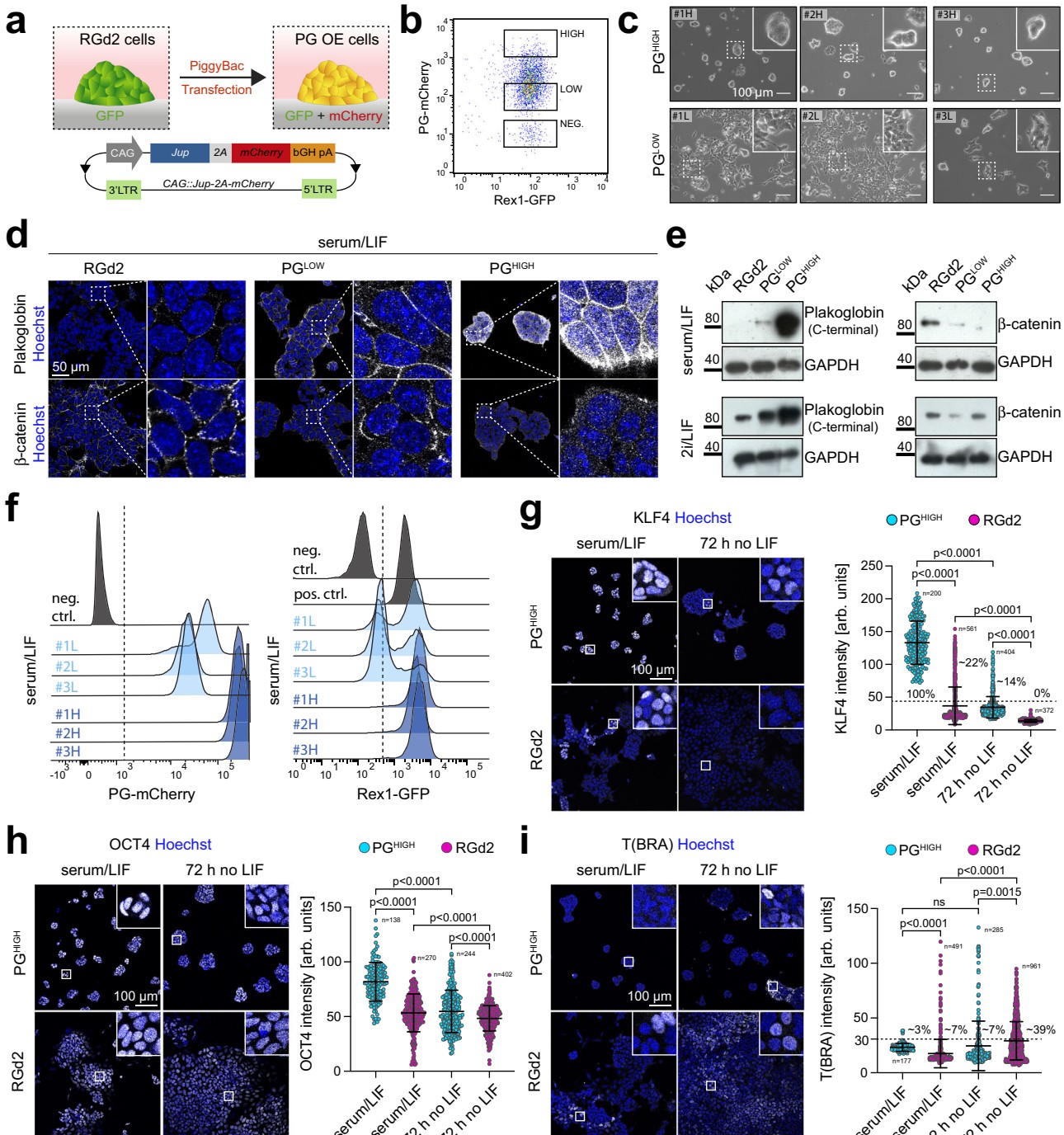

**Fig. 5 | Plakoglobin expression promotes naive pluripotency in metastable culture conditions. a** Schematic of the generation of Plakoglobin overexpressing RGd2 mESCs (PG OE). *Jup* (Plakoglobin) expression was placed under the control of a constitutively active CAG promoter and coupled to an mCherry fluorophore via a 2A self-cleaving peptide (*CAG::Jup-2A-mCherry*). This cell line simultaneously allowed monitoring of pluripotency via the Rex1-GFPd2 reporter and indicated the level of Plakoglobin via the mCherry signal. bGH pA: bovine growth hormone polyadenylation signal, LTR: long terminal repeats. **b** Flow cytometric analysis of PG OE mESCs. Single cells were sorted to generate clonal populations of mESCs with high (PG^HIGH) or low (PG^LOW) levels of Plakoglobin overexpression. **c** Phase contrast images of clonally expanded PG^LOW (#1L, #2L, #3L) and PG^HIGH (#1H, #2H, #3H) cells cultured in serum/LIF. Scale bar: 100 μm. Representative images for more than 10 passages. **d** Confocal immunofluorescence images of RGd2, PG^LOW#2, and PG^HIGH#3 cells stained for Plakoglobin (PG) and β-catenin. Scale bar: 50 μm. Representative of

3 independent experiments. **e** Western blot analysis for Plakoglobin and β-catenin in RGd2, PG^LOW, and PG^HIGH cells in serum/LIF and 2i/LIF. GAPDH was used as loading control. Representative of 3 independent experiments. **f** Flow cytometric analysis of the established clonal cell lines PG^LOW (#1L, #2L, #3L) and PG^HIGH (#1H, #2H, #3H) when cultured in serum/LIF. All PG^HIGH clones displayed a homogeneous Rex1-GFPd2 signal indicating acquisition of naive-like pluripotent state. **g–i** Confocal immunofluorescence images of RGd2 and PG^HIGH cells cultured in metastable pluripotent (serum/LIF) conditions and after 72 h of LIF withdrawal stained for the naive pluripotency marker KLF4 (**g**), the general pluripotency marker OCT4 (**h**), and the mesoderm marker T(BRA) (**i**). Scale bar: 100 μm. Bottom: Nuclear fluorophore intensity quantification. For KLF4 and T (BRA), positive cells (KLF4 threshold: 45 arb. units; T (BRA) threshold: 30 arb. units) are indicated in [%]. Error bars indicate the mean and standard deviations. *p*-values were determined by a two-tailed unpaired *t*-test with Welch's correction; 3 independent experiments.

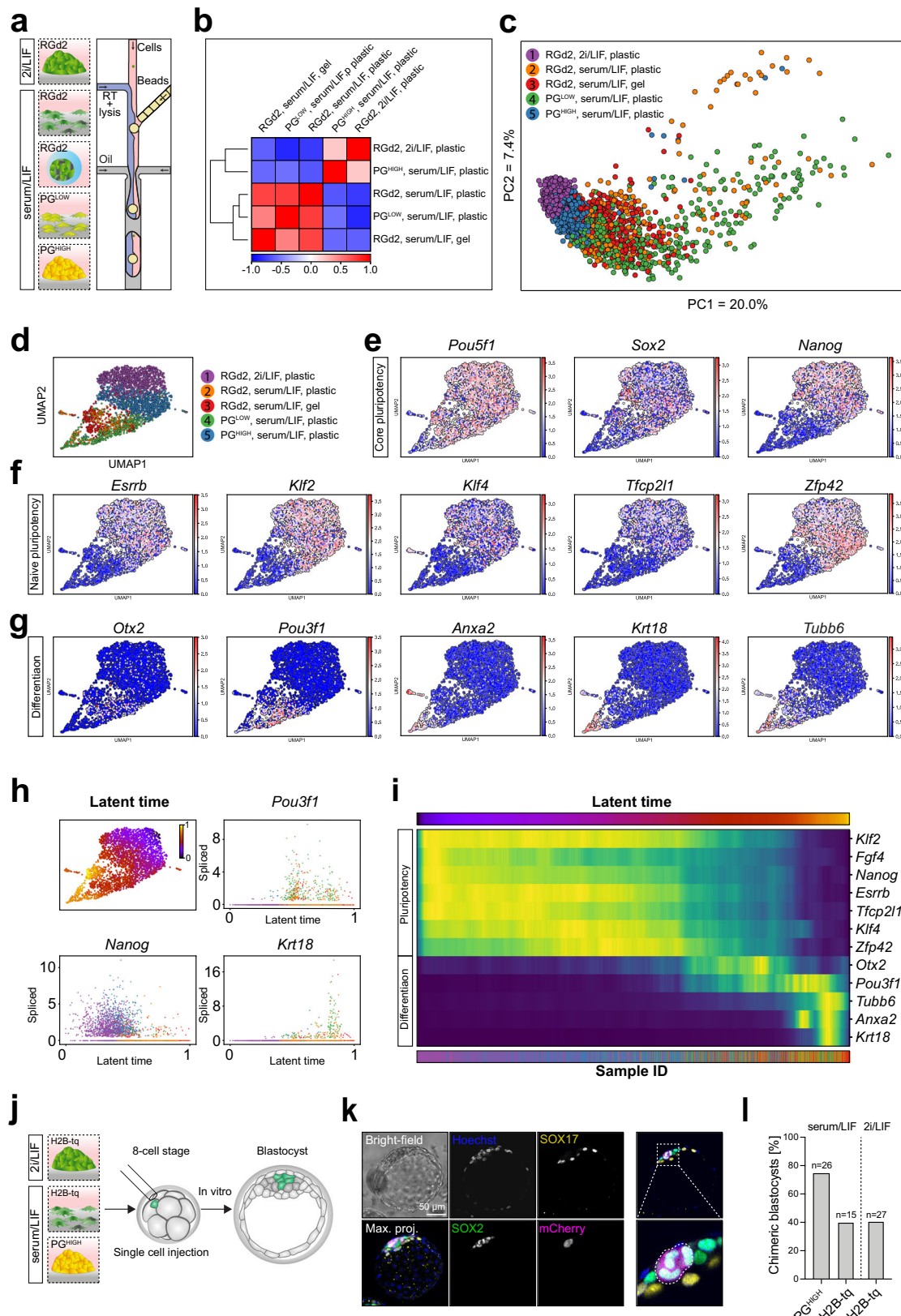

amounts of apparently truncated β-catenin in clone #2 (Figure S7e). PG^HIGH *Ctnnb1* KO cells remained KLF4 positive when cultured in 2i/LIF (Figure S7f). To challenge Plakoglobin's ability to maintain pluripotency, we exposed PG^HIGH *Ctnnb1* KO cells to either LIF, PD or CH alone (Figs. 7g–i, S7g). Remarkably, all PG^HIGH *Ctnnb1* KO clones maintained the dome-shaped morphology associated with naive

pluripotency even after 30 days of culture in LIF or PD (Fig. 7g). In LIF alone, Rex1-GFPd2 levels were unaffected, and all cells remained GFP+ (Fig. 7h). When exchanged from 2i/LIF to PD alone, PG^HIGH *Ctnnb1* KO underwent an initial loss of Rex1-GFPd2 signal which was recovered after several passages to ~75% (Fig. 7h). In contrast, a culture with the WNT agonist CH led to rapid loss of the Rex1-GFPd2 reporter and cells

**Fig. 6 | Single-cell sequencing elucidates plakoglobin-induced re-establishment of naive pluripotency. a** Schematic of the different samples (RGd2 in serum/LIF in microgels and on plastic, PG$^{low}$ and PG$^{high}$ cells in serum/LIF on plastic and naive control RGd2 cells in 2i/LIF on plastic) subjected to scRNA-seq analysis using the inDrop workflow. (RT: reverse transcriptase). **b** Pearson correlation heatmap of samples listed in (**a**). **c** Principal component (PC) analysis of samples listed in (**a**). **d** Uniform Manifold Approximation and Projection (UMAP) dimensional reduction plot of the scRNA-seq samples, for the maps of individual markers (in **e**–**g**). **e** Gene-expression values projected on the UMAP plot for core pluripotency markers (*Pou5f1, Sox2,* and *Nanog*). **f** Gene-expression values projected on the UMAP plot for naive pluripotency markers (*Esrrb, Klf2, Klf4, Tfcp2l1,* and *Zfp42*). **g** Gene-expression values projected on the UMAP plot for peri-implantation (*Otx2* and *Pou3f1*) and serum-induced differentiation markers (*Anxa2, Krt18,* and *Tubb6*). **h** (top left) Latent time computation generated using scVelo and projected on the UMAP underlining the continuum between naive pluripotency (latent time = 0) and

differentiation (latent time = 1). (top right) Single-cell spliced read counts arranged across the computed latent time for *Pou3f1* (top right), *Nanog* (bottom left), and *Krt18* (bottom right). **i** Heatmap representing the changes in gene expression across all cells arranged by latent time progression. **j** Schematic of single-cell injection into 8-cell stage embryos with subsequent in vitro culture until the blastocyst stage. **k** Confocal immunofluorescence image of a chimeric blastocyst that was injected at the 8-cell stage with a single serum/LIF cultured PG$^{HIGH}$ mESC. Blastocysts were stained for SOX2 (epiblast marker) and SOX17 (primitive endoderm marker). PG$^{HIGH}$ cells were identified by the PG-mCherry signal as shown in the merged image and highlighted by the white dotted line. Scale bar: 50 μm. **l** Chimeric blastocyst contribution efficiency of PG$^{HIGH}$ cells in serum/LIF (*N* = 26 embryos) and control H2B-tq cells in naive (2i/LIF; *N* = 27 embryos) and metastable pluripotent (serum/LIF; *N* = 15 embryos) conditions. Single cells were injected at the 8-cell stage followed by in vitro culture until blastocyst stage.

could not be maintained alive (Fig. 7h). Immunofluorescence analysis revealed that PG$^{HIGH}$ *Ctnnb1* KO cells remained positive for the naive pluripotency marker KLF4, albeit in a heterogenous fashion, even after 8 passages (16 days) in LIF alone. Sole supplementation with PD was not sufficient to maintain KLF4 expression (Fig. 7i). However, regardless of whether cultured in LIF or PD alone, all PG$^{HIGH}$ *Ctnnb1* KO cells remained positive for the general pluripotency marker OCT4 (Figure S7g). Next, we sought to test Plakoglobin's ability to prolong pluripotency in vivo. To this end, we aggregated 8-cell stage embryos with PG$^{HIGH}$ *Ctnnb1* KO or control mESCs and cultured them in vitro until E5.0 (Figure S7h). Both, control and PG$^{HIGH}$ *Ctnnb1* KO cells epithelialized and displayed OTX2 (early post-implantation) upregulation as expected (Figs. 7j, S7i). Strikingly, PG$^{HIGH}$ *Ctnnb1* KO, but not control cells, remained NANOG (naive pluripotency) positive despite having formed a monolayered epithelium and undergone lumenogenesis. In wild-type embryos, NANOG expression and epithelialization are found to be mutually exclusive[64,65]. Maintenance of NANOG in the epiblast epithelium with simultaneous expression of OTX2 therefore suggest a delayed exit from naive pluripotency in vivo regardless of exhibiting the morphology of a formative/primed pluripotent epiblast. Together, these data establish that the naive pluripotent circuitry is partially supported by Plakoglobin, independently of β-catenin.

## Discussion

It has become increasingly clear that cell behavior and developmental processes are not only regulated via biochemical signals but also respond to a plethora of mechanical signals. However, the interdependence of these levels of control remains underdetermined[66,67]. Conventional 2D plastic dishes do not emulate the complex in vivo environment and are unsuitable to decipher cell signaling in 3D. New experimental formats are needed to shed light on how gene regulatory networks, biochemical and biomechanical signals are integrated within cells. Here we used a microgel system to encapsulate mouse embryonic stem cells into spherical 3D agarose scaffolds. Consistent with previous studies, we found the soft, non-degradable 3D matrix to be beneficial for pluripotency[21,25,68,69].

The interpretation of RNA-seq data on microgel-encapsulated cells revealed Plakoglobin, a vertebrate homolog of β-catenin and a crucial factor for embryonic development, as one of the most differentially upregulated genes[40,70]. Indeed, both proteins are known to be crucial for mouse embryonic development[70–72]. β-catenin is a well-known mechanotransducer in several species and cell types[4,73–76], while the recruitment of Plakoglobin to adherens junctions under tension in *Xenopus* mesendoderm cells is the only documented example that suggest a role in mechanotransduction[77]. To our knowledge, little is known about Plakoglobin's ability to react to biophysical signals in mammals—in particular throughout pluripotency. Remarkably, we observed that Plakoglobin was exclusively upregulated in microgels but not in hanging drops, suspension

culture (3D environment without confinement) or on soft 2D gels. We therefore suggest that gel-mediated confinement acts as an upstream mechanical signal for Plakoglobin expression. Notably, Plakoglobin expression in vivo correlates with the expansion of the blastocyst cavity and the increase in pressure within the blastocyst, an expression pattern we observed in both mouse and human embryos[31,32,78–80]. Yet, whether there is a mechanoresponsive component of Plakoglobin regulation in vivo remains to be investigated. Simultaneously with the expression of Plakoglobin in the pre-implantation epiblast, naive pluripotency is established[9,78]. This observation and the abrupt disappearance of Plakoglobin upon implantation (and transition into primed pluripotency) suggest a role for Plakoglobin in specifically regulating naive pluripotency. Plakoglobin, unlike β-catenin, is known to activate TCF/LEF-mediated transcription only weakly[81,82] and endogenous levels of Plakoglobin cannot rescue β-catenin null embryos during gastrulation[71,72]. Plakoglobin has been shown to delay differentiation[81] but its ability to maintain and regulate pluripotency is not known. Here, comprehensive single-cell sequencing and blastocyst injections confirmed that Plakoglobin overexpressing cells re-acquire naive-like pluripotency under the metastable pluripotent culture conditions of serum/LIF. In *Xenopus* it has been suggested that Plakoglobin could act indirectly through β-catenin by releasing it from the membrane and saturating its degradation machinery whilst another study found that in mammalian cells Plakoglobin expression led to the degradation of β-catenin[83]. Here, XAV treatment (blocking nuclear translocation of β-catenin) or a β-catenin knockout in Plakoglobin-overexpressing cells does not compromise their ability to maintain the pluripotency network. This suggests, that Plakoglobin can act independently of β-catenin in supporting pluripotency. Whether Plakoglobin's beneficial effect on pluripotency acts in a redundant fashion to β-catenin, by alleviating repressive effects of TCF7L1 on the pluripotency network, remains to be investigated[16,17,84,85].

In summary, the agarose microgel format allowed us to investigate the effects of microenvironmental confinement on pluripotent mESCs. We identified Plakoglobin (*Jup*) expression to be confinement-induced. This mechanism that would have been missed under conventional cell culture conditions, suggesting that this new experimental format can be of analytical value. Furthermore, owing to Plakoglobin's expression in human and mouse pre-implantation embryos—in particular the expression in the epiblast, and its ability to reinstate naive pluripotency in mESCs – led us to propose Plakoglobin as an evolutionarily conserved, mechanoresponsive regulator of naive pluripotency. In the future, our approach of using agarose microgels may be instructive to elucidate confinement-mediated signaling cascades beyond pluripotent stem cells, e.g. in developmental processes such as gastrulation and diseases including cancer—as all of these have been postulated to involve mechanosensitive signaling pathways[86,87].

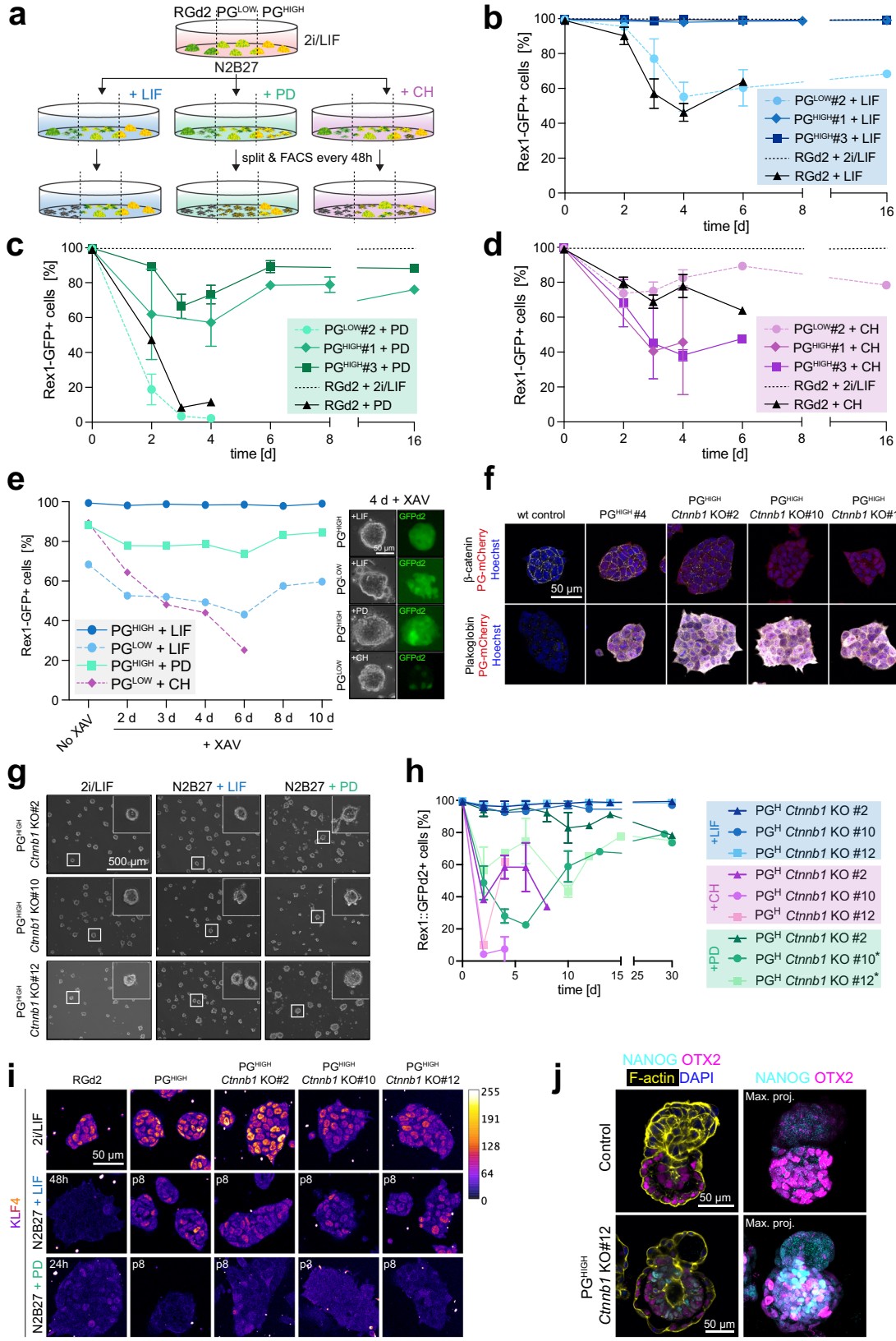

## Methods

### Cell culture

**Mouse embryonic stem cells.** The mouse embryonic stem cell lines E14 and Rex1GFPd2 (RGd2) were a kind gift from Austin Smith's laboratory at the University of Cambridge. They were generally cultured in 2i/LIF medium[12], which consists of N2B27 medium (1:1 DMEM/F-12

and Neurobasal media, N2 supplement[88] (Cambridge Stem Cell Institute), B-27 supplement[89] (gibco™), 0.11 % BSA fraction v (gibco™), 0.1% sodium bicarbonate (gibco™), 12.5 μg/mL human insulin recombinat zinc (gibco™), 50 μM 2-mercaptoethanol (gibco™), 2 mM L-glutamine (gibco™), 1x penicillin-streptomycin (gibco™)) and is supplemented with 1 μM PD0325901 (Cambridge Stem Cell Institute), 3 μM CHIR99021

**Fig. 7 | Plakoglobin sustains pluripotency independently of β-catenin.**
**a** Experimental layout of culture conditions to assess Plakoglobin's potential to maintain naive pluripotency. PD (MEK inhibitor) CH (WNT agonist). **b**–**d** Flow cytometric analysis of the Rex1-GFPd2 reporter in the PG^HIGH (clone #1 and #3), PG^LOW, and RGd2 cell lines. Cells were cultured up to 16 days with the sole supplementation of PD (**b**), LIF (**c**), or CH (**d**). The error bars show the standard deviations of 3 independent experiments. **e** Flow cytometric analysis of the Rex1-GFPd2 reporter in the surviving cell lines (established from **b**–**d**) after treatment with the WNT-signaling antagonist XAV. **f** Confocal immunofluorescence images of PG^HIGH, PG^HIGH *Ctnnb1* KO (clone #2, #10, and #12) and wild type control cells in 2i/LIF stained for β-catenin and Plakoglobin. Scale bar: 10 μm. Representative of 3 independent experiments. **g** Phase contrast images of PG^HIGH *Ctnnb1* KO (clone #2, #10, and #12) cells cultured in 2i/LIF or with the single supplementation of LIF, PD for

30 days. Scale bar: 500 μm. **h** Flow cytometric analysis of the Rex1-GFPd2 reporter in PG^HIGH *Ctnnb1* KO (clone #2, #10, and #12) cells when cultured with the sole supplementation of LIF, PD, or CH. Cells were analyzed every 48 h for up to 30 days. The error bars show the standard deviations of 3 independent experiments. **i** Confocal immunofluorescence images of RGd2 and PG^HIGH *Ctnnb1* KO (clone #2, #10, and #12) cells cultured in 2i/LIF, LIF-only, and PD-only stained for the naive pluripotency marker KLF4. Scale bar: 50 μm. Representative of 3 independent experiments. **j** Representative confocal immunofluorescence images of mouse embryo chimeras. 8-cell embryos were aggregated with 3–5 control (N = 19 embryos) or PG^HIGH *Ctnnb1* KO#12 (N = 24 embryos) cells and cultured in vitro until a post-implantation-like state (see "Methods") and stained for NANOG and OTX2. Scale bar 50 μm.

(Cambridge Stem Cell Institute), and 20 ng/mL leukemia inhibitor factor (Cambridge Stem Cell Institute). For a certain set of experiments, mESCs were cultured in serum/LIF medium (GMEM (Merck) supplemented with 10% fetal calf serum (FCS) (Merck), 1x non-essential amino acids (NEAA) (Life Technologies), 1 mM sodium pyruvate (Life Technologies) and 1 mM L-Glutamine (Life Technologies)). Cells were maintained on gelatin-coated (0.1%) culture dishes in a 7% $CO_2$ humidified incubator at 37 °C and medium was replaced every other day. Cells were passaged every 2 to 3 days at $80 \times 10^4$ cells/6-well (~8000 cells/cm²). To passage the cells or dissociate the cells into single cells for further experimental procedures the cells were treated as follows: the medium was aspirated, and cells were treated with Accutase® at 37 °C until cells detached from the dish. Colonies were dissociated into single cells by gentle pipetting of the detached cells against the tissue culture dish. Afterward, the Accutase® was diluted with 10x volume of wash medium (DMEM-F12 + 1% BSA) and cell were centrifuged for 3 min at 300 RCF. Finally, the supernatant was discarded, cells were resuspended and replated in the appropriate pre-warmed medium.

**Epiblast-derived stem cells.** Mouse epiblast-derived stem cells (EpiSCs) reporter were a kind gift from Austin Smith's laboratory at the University of Cambridge. EpiSCs were cultured in N2B27 media supplemented with 12 ng/mL bFGF (Cambridge Stem Cell Institute), 20 ng/mL activin A (Cambridge Stem Cell Institute), and 2 μM XAV 939 (Tocris) on fibronectin (10 μg/mL, Merck) coated 6-well plates. They were passaged every other day by Accutase® treatment and seeded at a cell density of ~10,000 cells/cm².

**Human pluripotent stem cells.** The conventional human pluripotent cell line H9[90] was propagated on Geltrex coated 6-well dishes in Essential 8 (E8) medium[91] made in-house. Briefly, DMEM/F12 was supplemented with 64 mg/L L-ascorbic acid-2-phosphate magnesium, 100 μg/L FGF2, 19.4 mg/L insulin, 14 μg/L sodium selenium, 543 mg/L NaHCO₃, 10.7 mg/L transferrin, and 2 μg/L TGFβ1 or 100 μg/L Nodal. The human naive derived ES cell line HNES1[53] was routinely maintained on inactivated mouse embryonic fibroblast (iMEF) feeder cells in PXGL medium[92,93]. Briefly, N2B27 was supplemented with 1 μM PD0325901, 2 μM XAV939, 2 μM Gö6983, and 10 ng/mL human LIF. For immunofluorescent staining, HNES1 cells were plated on laminin-coated dishes to circumvent the use of iMEFs. For capacitation, naive HNES1 cells were transferred to N2B27 with 2 μM XAV939 on iMEFs for 11 days, passaged at days 6 and 11. Subsequently, capacitated cells were cultivated on iMEFs in E8. Naive and conventional human ES cells were fed every day and passaged via Accutase treatment every 3 to 5 days and cultured in 5% $O_2$ and 7% $CO_2$ in a humidified incubator at 37 °C. After passaging, 10 μM ROCK inhibitor (Y-27632, 688000, Millipore) were added to the culture for 24 h.

### Embryo studies

**Mouse embryo dissection and culture.** All mice used were intercrosses of strain CD1 and obtained through natural mating. The sex of

embryos and the ages of mice using mating were not considered in this study. The mice were maintained in a state-of-the-art biofacility with daily health checks carried out by dedicated trained staff. The mice were maintained on a lighting regime of 12 h:12 h light:dark with food and water supplied ad libitum. This research has been regulated under the Animals (Scientific Procedures) Act 1986 Amendment Regulations 2012 following ethical review by the University of Cambridge Animal Welfare and Ethical Review Body (AWERB). Use of animals in this project was approved by the ethical review committee for the University of Cambridge, and relevant Home Office licences are in place. Embryonic staging assumed, that mating occurred at midnight. Hence, embryos were staged at E0.5 and noon of the following day. Pregnant females were killed on day 2.5, day 3.5, and day 4.5 post coitum (E2.5, E.3.5, and E4.5) by cervical dislocation. Oviduct and uterus were dissected and flushed with M2 medium (Merck) using Leica M165C stereo microscope. Post-implantation E5.5 and E6.5 embryos were carefully dissected from the implantation sites within the uterus and washed in M2 medium before fixation and subsequent processing.

**Embryo injections for chimera generation.** Oviduct and uterus were dissected and flushed with M2 medium (Merck) using Leica M165C stereo microscope. For (single) cell injections, mESCs grown in the appropriate conditions were dissociated by Accutase® treatment for 5 min at 37 °C. Cells were then resuspended in washing buffer (9.5 mL, DMEM/F-12 + 5% BSA) and centrifuged for 3 min at 300 RCF. Subsequently, cells were resuspended in their appropriate medium (2i/LIF or serum/LIF) and stored on ice until injection. mESCs were loaded into a microinjection pipette and injected into the perivitelline space of 8-cell stage embryos via laser-generated perforation of the zona pellucida using the XYClone® (Hamilton Thorne Biosciences). Subsequently, injected embryos were cultured in ORIGIO® Blast™ medium (Origio) at 37 °C and 7% $CO_2$ for 48 h, the equivalent of the E4.5 blastocysts.

**Culture of mouse embryo chimeras to early post-implantation stages.** Embryos from CD1 females (Charles River) were collected at E2.5. Zona pellucida was removed through short incubation in acidic Tyrode's solution (Sigma-Aldrich, T1788) and the embryos washed in KSOM (Sigma-Aldrich, MR-106-D) and transferred to individual KSOM drops and aggregated with small clumps of mESCs (3–5 cells) and cultured overnight at 37%, 5% $CO_2$ in a humidified atmosphere. Following day, compacted and cavitated embryos were transferred to in vitro culture medium adjusted from Ma et al.[94] as published previously[95] supplemented with 10% FBS (v/v). Next day in the evening, the mural trophectoderm was cut following initiation of implantation and the remaining embryo (polar trophectoderm, epiblast, and primitive endoderm) transferred to drops of in vitro culture medium supplemented with 20% FBS (v/v). 24 h later, the embryos were fixed in 4% PFA in PBS for 20 min.

**Immunofluorescence staining of embryos.** Initially, the zona pellucida was removed using acid Tyrode's solution. Subsequently, embryos

were fixed with 4% PFA for 20 min at room temperature. After one wash in PBS supplemented with 3 mg/mL polyvinylpyrrolidone (PBS/PVP), embryos were permeabilized for ~30 min in PBS/PVP with 0.25% Triton X-100. Afterward, embryos were blocked in PBS containing 0.1% BSA, 0.01% Tween 20, and 2% donkey serum (blocking buffer). Primary antibodies (see Supplementary Information) were appropriately diluted in blocking buffer and incubated at 4 °C overnight. Subsequently, embryos were washed 3x in blocking buffer for at least 15 min each time before incubation with the secondary antibodies for 2 h at room temperature and in the dark. Subsequently, embryos were washed in blocking buffer. Imagining was performed by placing the embryos in 2 μL droplets of water in a microscopy dish covered with mineral oil.

**Human embryo culture, immunofluorescence staining and ethics statement.** Use of human embryos in this work has been approved by the Multi-Centre Research Ethics Committee, approval O4/MRE03/44 and approved by the Human Embryology & Fertilization Authority of the United Kingdom, research license R0178. Frozen embryos, surplus to requirement, were donated with informed consent by couples undergoing in vitro fertility treatment. Embryos were thawed using EmbryoThaw™ Kit (EMF40_T, Fertipro) according to the manufacturer's instructions. 4-cell stage embryos were cultured briefly in Cleave™ (Origio). Blastocysts were thawed directly into N2B27 medium (SCS-SF-NB-02, Stem Cell Sciences) and cultured to the desired stage in 50 μL drops of medium under mineral oil (MINOIL500, Fertipro) in a humidified atmosphere at 5% $O_2$, 7% $CO_2$ and 37 °C. Embryos were processed for immunohistochemistry by fixation in 4% paraformaldehyde in PBS for 15 min, rinsing in PBS containing 3 mg/mL polyvinylpyrrolidone (PBS/PVP; P0930, Sigma), permeabilized in PBS/PVP containing 0.25% Triton X-100 (23,472-9, Sigma) for 30 min and blocked in buffer comprising PBS, 0.1% BSA, 0.01% Tween 20 (P1379, Sigma), and 2% donkey serum. Primary antibodies (see Supplementary) were diluted in blocking buffer and embryos incubated in antibody solution at 4 °C overnight. They were rinsed three times in blocking buffer for 15 min each, and incubated in secondary antibody solution for 1 h at room temperature. Embryos were rinsed three times in blocking buffer, incubated briefly in increasing concentrations of Vectashield (H-1200, Vector Labs, Peterborough, UK) before mounting on glass slides in small drops of concentrated Vectashield with DAPI, and subsequently sealed with nail varnish.

**Immunofluorescence staining of marmoset embryo cryosections.** Carnegie stage 6 marmoset cryosections originated from embryos of a recently published[52] study. Briefly, slides were thawed at room temperature and fixed for 8 min in 4% PFA/PBS solution (15714S, Electron Microscopy Sciences/Thermo Fisher Scientific), and then washed three times with PBS. Permeabilization was performed in 0.25% Triton X-100 (13444259, Thermo Fisher Scientific) in 0.3% polyvinyl pyrrolidone/ PBS (Thermo Fisher Scientific) for 30 min at room temperature. The slides were rinsed 3 times with PBS and incubated for 30 min in blocking buffer (2% donkey serum (116-4101, Thermo Fisher Scientific), 0.1% bovine serum albumin (BSA; A9418, Sigma-Aldrich), 0.01% Tween 20 (BP337-100, Thermo Fisher Scientific) in PBS) at room temperature.

### Bulk and scRNA-seq of mouse embryonic stem cells
**Bulk RNA library preparations and sequencing.** For each triplicate condition, a total of 1000 cells were dispensed in single wells and processed using the Smart-seq2 protocol[96]. The amount of PCR cycles for the second-strand generation step were adjusted to 12 cycles to account for the increased input material. The unfragmented libraries were quality-controlled using a Bioanalyzer 2100 HS kit (Agilent). The libraries were then processed for tagmentation, using 1 ng of amplified cDNA as input, and 8 cycles were used for the dual-indexing PCR. The DNA molarity of the libraries after AMPureXP (Beckman Coulter) clean-up was assessed using a Bioanalyzer 2100 HS and a Qubit HS (Thermo

Fischer) to estimate size distribution and DNA concentration respectively. Finally, the libraries were pooled at equimolar ratios and sequenced using a MiSeq v2 300 cycle kit (Illumina) (DNA Sequencing Facility, Dept. of Biochemistry).

**Bulk RNA-seq data analysis.** The fastq files were quality inspected using fastQC. The STAR aligner[97] was used for mapping each de-multiplexed paired-end file to a mm10 reference genome with a Gencode M12 gtf annotation file. Counting was achieved using feature-Counts from the subread package[98], and multi-mapped reads were discarded. Count tables were then imported in DESeq2[99] for differential expression analysis between the triplicate conditions and Volcano plots were obtained using the EnhancedVolcano tool[100]. For GSEA analysis, we used the stat parameter provided from the DESeq2 gene expression analysis as an input for the WebGESTALT tool[101].

**inDrop single-cell RNA-seq library preparation.** Cells were resuspended at a final concentration of 120,000 cells/mL in PBS supplemented with 15% OptiPrep™ and libraries were prepared according to the inDrop protocol[102] using the v3 barcoding configuration[103]. Briefly, each cell sample was encapsulated in water-in-oil emulsions containing a barcoded polyacrylamide microgel, a lysis buffer, and reverse transcriptase mix. Each sample was collected in a single fraction containing approximately a 1000 cells and libraries were processed according to the protocol. The bead-bound barcode was released from the hydrogel using UV exposure and the droplets were incubated at 50 °C for 2 h followed by 70 °C for 20 min. The libraries were then processed for second-strand synthesis and amplification using in vitro transcription. After reverse-transcription, the libraries were amplified using limited-cycle indexing PCR. The final libraries were quality controlled using a BioAnalyzer HS kit (Agilent). Samples were pooled at equimolar ratios using both the BioAnalyzer HS and Qubit HS (ThermoFisher) metrics for size distribution and DNA concentration quantification. The libraries were sequenced on a Nextseq 75-cycle High Output kit (Illumina) in stand-alone mode using a 5% PhiX spike-in as an internal control.

**Single-cell RNA-seq data analysis.** The BCL files were converted to fastq files using the bcl2fastq script from Illumina. The read files were quality controlled using fastQC and de-multiplexed using Pheniqs[104]. The de-multiplexed files were used as an input for the zUMIs pipeline[105] and mapped to a mouse GRCm38 reference genome with GRCm38.99 gtf annotation. The obtained aggregated intronic and exonic count matrices were then processed with the Scanpy tool[106]. Cells that had a low (<1%) or high fraction (>12%) of counts mapped to mitochondrial genes were excluded. In addition, cells were filtered on the detected number of genes (between 800 and 3000) and transcript counts (between 1500 and 50,000). The data objects were then normalized, and transcript counts were regressed out. After scaling and principal component analysis, a dimensional reduction plot was obtained by computing a two-dimensional Uniform Manifold Approximation and Projection (UMAP) was achieved using the scanpy.tl.umap() function. Clustering was achieved using the leiden algorithm (sc.tl.leiden()). Dynamical modeling of RNA velocity was performed with the scVelo tool[58] using the velocyto loom files[57] obtained via the zUMIs pipeline as an input. Latent times were computed using the scVelo package.

### Microscopy
**Confocal microscopy.** Imaging was performed with inverted Leica TCS SP8 confocal microscopes using 40x/1.30 and 20x/0.75 objectives. Fluorophores were excited with a 405 nm, a 488 nm, a 552 nm, and a 638 nm laser. Raw data were analyzed with the open-source software ImageJ/Fiji[107].

**Atomic force microscopy.** AFM force measurements were performed on a JPK CellHesion 200 AFM (Bruker) using an Arrow TL1 cantilevers

(NanoWorld, manufacturer-supplied nominal spring constant value is 0.03) onto which a 37 μm polystyrene bead (microParticles) had been glued. For sample preparation low melting agarose (lonza) was mixed with PBS (1.5% w/v), dissolved at 70 °C, and subsequently cooled down to 37 °C. 2 mL were poured into TTP™ dishes (P/N 93040), cooled down in the fridge overnight and PBS was either added directly before the measurement or 1.5 h before. For the measurements set points of up to 60 nN were chosen and the speed of the Z scanner was 5 mm/s. Each gel was measured in 16 locations. The resulting force-distance curves were then analyzed in the JPK Data Processing software, where the Hertz Model was applied to calculate the Young's Modulus, assuming Poisson's ratio to be 0.5. Care was taken that the indentation remained below one-third of the bead radius.

### Microfluidics

**Fabrication of master molds.** The microfluidic chips were designed with the computer-aided design (CAD) software (DraftSight) and were produced using well-established soft-lithography protocols[108]. Briefly, chip designs were cut out on a photo mask which was then used to fabricate a silicon wafer master mold in a clean room.

**Microfluidic chip production.** The silicon master molds were placed in a petri dish and filled with SYLGARD™ 184 silicone elastomer base and curing agent (Merck) in a 10:1 (w/w) ratio. After degassing, the PDMS was polymerized at 65 °C overnight and carefully removed from the master mold. The in- and outlets for the chip were generated with a 1 mm biopsy puncher (pmf medical). After thorough cleaning with 2-propanol and compressed air, the PDMS and microscope slides were placed into a low-pressure oxygen plasma generator. A vacuum (0 mbar) was generated, and oxygen plasma introduced for 12 s. The PDMS was pressed onto the microscope slides to form covalent bonds, the channels where coated/treated with 1% (v/v) Trichloro(1H,1H,2H,2H-perfluorooctyl)silane (Merck) in HFE-7500 (Fluorochem) and the chips were placed on a 50 °C hot plate for 30 min. Until use, chips were stored in a sterile container.

**Encapsulation.** Low melting-point agarose (Lonza) was mixed with PBS, heated up to 70 °C to form a 3% (w/v) solution and kept at 37 °C until the encapsulation. A single-cell suspension was mixed gently but thoroughly with the agarose solution (1:1 ratio) and carefully aspirated into a polyethene tubing (Portex®) connected to a glass syringe (SEG Analytical Science). Another syringe was filled with HFE-7500 oil (Fluorochem) and Pico-Surf surfactant (Sphere Fluidics) with the final concentration of 0.3%. The Syringes were connected to the microfluidic chip and automated pumps (CETONI) injected both the oil-surfactant solution and the agarose-cell suspension through different channels. Agarose droplets were formed and collected on ice to allow polymerization of the agarose. Using liquid-liquid extraction the microgels were demulsified with 1H,1H,2H,2H-perfluoro-1-octanol (AlfaAesar) and culture medium.

**Cell culture in agarose microgels.** Generally, agarose microgel encapsulated mESCs were cultured in 2 mL 2i/LIF (naive pluripotent), serum/LIF (pluripotent), or N2B27 (differentiation) medium as static suspension culture in 6-well plates. For culture longer than 48 h, wells were coated for 15 min with 0.1% gelatin to enable escaping cells to adhere to the plastic and not influence downstream analysis. If cultured for more than 48 h, media were exchanged every day by collecting the media and microgels, centrifugation at 200 RCF for 3 min, careful aspiration of the media and transfer of the microgels into a new 6-well plate with fresh media.

**Cell recovery from microgels.** For the recovery of mESCs from agarose microgel, the cell culture medium including the microgels was collected in a tube and gently centrifuged at 200 RCF for 3 min. The

medium was aspirated until only 1 mL was remaining and 2 μL Agarase (0.5 U/μL, Thermo Fischer) was added and gently mixed. They were incubated at 37 °C until the microgels were dissolved and the cell colonies released (~5–10 min). The cell colonies were then either used for further experiments, after the transfer into new medium, or they were pelleted at 300 RCF for 3 min and treated with Accutase® to generate single cell suspension for example for flow cytometry analysis.

### Molecular biology

**Cloning of the *Jup-T2A-mCherry* plasmid.** The transposon for the genomic integration of *Jup* using the PiggyBac system was constructed as follows. The *Jup* cDNA was recovered from the bulk RNA-seq experiment cDNA pool and amplified for downstream vector assembly. The PCR-amplified *Jup* cDNA (primer pair: F_Jup-cDNA and R_Jup-cDNA) was inserted into the previously described pPB-CAG vector[87] together with an mCherry gene via Gibson assembly. For bi-cistronic expression under the control of the CAG promoter, mCherry was linked to Jup via the ribosome-skipping 2A sequence of *thosea asigna* virus (T2A). The final *PB-CAG-Jup-T2A-mCherry* construct was validated by Sanger sequencing (DNA Sequencing Facility, Dept. of Biochemistry).

**Generation of Plakoglobin overexpressing RGd2 mESCs.** To generate the *Jup* overexpressing cell line, the PiggyBac transposon system[87] was used. Briefly, 10 μL Lipofectamine 2000 were mixed with 300 μL 2i/LIF medium. Separately, 1.2 μL *PB Jup-T2A-mCherry* construct (700 ng/mL, 0.8 μg) with 1.6 μL PBase (500 ng/mL, 0.75 μg) and 150 μL 2i/LIF medium. Then, 150 μL diluted DNA were mixed with 300 μL diluted Lipofectamine 2000 and incubated for 5 min at room temperature. Subsequently, ~5 × 10⁵ cells were resuspended in 450 μL of DNA/Lipofectamine 2000 and incubated for 10 min at 37 °C. Then, cells were plated and stably transfected cells FACS-sorted based on their mCherry signal.

**Ctnnb1 knock-out via CRISPR Cas9.** β-catenin knockout cells were generated via transient transfection with a CRISPR/Cas9 *Ctnnb1* KO plasmid according to the manufacturer's instructions (Santa Cruz, sc-419477).

*Ctnnb1* was targeted by a mixture of the following 3 gRNAs:
1: ATGAGCAGCGTCAAACTGCG
2: AGCTACTTGCTCTTGCGTGA
3: AAAATGGCAGTGCGCCTAGC

For knock-out validation, genomic DNA of grown clones was harvested using QuickExtract™ DNA Extraction Solution (*Lucigen*) following the manufacturer's instructions. The solution was directly used as a template for PCR amplification using Q5® High-Fidelity 2X Master Mix (*NEB*) (primer pair: F_Ctnnb1_Exon3to6 and R_Ctnnb1_Exon3to6). PCR products were resolved by agarose gel electrophoresis and relevant bands extracted and purified for Sanger sequencing (DNA Sequencing Facility, Dept. of Biochemistry) to identify genome edit at targeted sites.

### Cell analysis

**Hanging drop culture.** A single-cell suspension with either 333 cells/mL or 33 cells/mL was generated in serial dilution with 2i/LIF medium with prior Accutase® treatment of mESCs. Subsequently, 30 μL drops (containing ~10 cells or ~1 cell) were distributed onto the inside of a 15 cm petri dish and the lid was placed onto the PBS filled dish, so that the drops were hanging upside down in the lid without touching the PBS.

**Clonogenicity assay.** mESCs cultured in 2i/LIF or serum/LIF were either cultured on tissue culture plastic or encapsulated into agarose microgels. After 48 h the microgels were dissolved with Agarase (Thermo Fisher Scientific) and single-cell suspensions were generated with Accutase®, both from cells on tissue culture plastic and microgels. 1000 cells were seeded in 2i/LIF on gelatin-coated 6-wells (cell density of ~10⁴ cells/cm²). Surviving colonies were counted after 48 h.

**Flow cytometry and cell sorting.** The fluorescent cell reporters Rex1-GFPd2 and PG-mCherry were analyzed with either the LSRFortessa™ (BD Biosciences) or the Attune™ NxT (Thermo Fisher Scientific) flow cytometer. Single-cell suspensions were generated by Accutase® treatment. The cells were washed with DMEM-F12 + 5% BSA, resuspended in PBS or medium, and kept on ice until the measurement. The data were analyzed with the software FlowJo (BD Biosciences).

**Fluorescence-activated cell sorting.** Bulk and single-cell fluorescence-activated cell sorting (FACS) was performed at the Cambridge Stem Cell Institute Flow Facility on a MoFlow XDP cell sorter (Beckmann Coulter) and the Flow Cytometry School of the biological sciences on the BD FACSAria™ IIu Cell Sorter (BD Biosciences). The cells were treated with Accutase®, washed with DMEM-F12 + 5% BSA, resuspended in cell media, and stored on ice until FACS. Bulk sorts were collected in a collection tube and washed after the sorting before culturing the cells in a gelatin-coated 6-well plate. Single cells were directly sorted into gelatin-coated 96-well plates.

**Immunofluorescence staining of cells.** Cells were fixed for 10 min with 4% formaldehyde solution (Thermo Fisher Scientific) in PBS. After washing with PBS, cells were permeabilized for 15 min with 0.25% Triton X-100 in PBS. Subsequently, cells were blocked for at least 2 h with 3% BSA in PBS blocking solution. Primary antibody (see Supplementary Information) incubation was performed at desired concentration at 4 °C, overnight. Cells were then washed with blocking buffer and incubated with the secondary antibody for 2 h at room temperature. Afterward, cells were washed with blocking buffer and stained with 1 μg/mL Hoechst.

**Western blot analysis.** Protein lysates were generated by treating cells with lysis buffer (1x RIPA (Abcam), 1x cOmplete mini protease inhibitor (Roche)), followed by incubation on ice and centrifugation (13523 RCF, 20 min at 4 °C). The supernatants were collected and either used directly or stored at −80 °C. Protein concentrations were measured with the Qubit™ Protein Assay Kit (Invitrogen) and diluted to a final concentration of 1.25 μg/μL. Denaturing SDS page with NuPAGE 4−12% Bis-Tris gels (Invitrogen) was performed at 150 V in MOPS buffer (Invitrogen), followed by protein transfer onto Amersham™ Protran™ 0.45 μm NC membranes (GE Healthcare Life science) at 100 V for 90 min in transfer buffer (25 mM Tris, 192 mM Glycine). Membranes were blocked with filtered 5% milk in TBS-T buffer (50 mM Tris, 150 mM NaCl, 1% Tween-20, pH 7.6) followed by primary antibody (see Supplementary Information) incubation at 4 °C overnight at. The membranes were washed in TBS-T prior HRP coupled secondary antibody incubation (1:10,000) at room temperature followed by further washing steps. For detection Amersham ECL Prime Western Blotting Detection Reagents (Cytiva) and RX X-ray film (Fuji) were used according to manufactures protocol.

### Reporting summary
Further information on research design is available in the Nature Portfolio Reporting Summary linked to this article.

## Data availability
The datasets generated during and/or analyzed during the current study are available in the GEO data repository under accession number GSE197643. Source data are provided with this paper.

## Code availability
The scripts used for RNA-seq analysis can be found in Github [https://github.com/droplet-lab/Plakoglobin/] and in Zenodo [https://doi.org/10.5281/zenodo.7913104].

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

## Acknowledgements

T.N.K. and M.H. received scholarship support from AstraZeneca, J.D.J. from the BBSRC, A.L.E. from the Cambridge Trusts and the EU H2020 Marie Curie ITN MMBio and Ka.F. from the MRC and St. John's College Cambridge. This work was supported by the Wellcome Trust (WT108438/C/15/Z and WT RG89228). F.H. is an H2020 ERC Advanced Investigator (695669), T.E.B. is a Wellcome Trust and Royal Society Sir Henry Dale Fellow. We are grateful to Dr. Geraldine M. Jowett for critical reading and constructive comments on the manuscript. We further thank Alisa Maring regarding her expertise in Western blotting. Finally, we thank Joana Cerveira the Flow Cytometry Facility Manager of the School of the Biological Sciences at the University of Cambridge.

## Author contributions

T.N.K. and F.H. conceived and initiated the study. T.N.K. and A.L.E. carried out experiments across the entire range of approaches, analyzed data, prepared figures, and organized the preparation of the manuscript. J.D.J. prepared the inDrop and bulk sequencing libraries and analyzed transcriptomic data. A.Y. and An.W. performed the mouse embryo injections and aggregations. J.N. holds the required HFEA and Home Office licenses and carried out the

human embryo experiments. M.H. and Ka.F. assisted with generation of cell lines. Al.W. and Kr.F. implemented AFM measurements and their analysis. Cl.M., E.M.S., C.R., and S.B. contributed exploratory data and Ca.M. and K.C. experimental approaches. T.N.K., T.E.B., and F.H. wrote the manuscript with input from all authors. T.E.B. and F.H. supervised the work.

## Competing interests

The authors declare no competing interests.
