## [Peer Review File · Nature Communications]

Plakoglobin is a mechanoresponsive regulator of naïve pluripotencyREVIEWER COMMENTS

Please note that Reviewer #3's full comments were submitted in a file, and are included in this email as an attachment.

Reviewer #1 (Remarks to the Author):

Overall, this is an excellent study, the findings are highly interesting and well documented, and I support the publication.

The data is beautifully presented and the article is well written.

Despite a substantial body of work defining the chemical and cytokine-mediated signals that control pluripotency transcriptional programs, biomechanical cues and mechanoresponsive genes as emerging key players are ill-defined. The authors of this paper elegantly utilise an encapsulated microgel platform in which mESCs grow under self-renewing or differentiating conditions and display morphological development. Through this work, they identify Plakoglobin (a vertebrate homolog of β -catenin) as a mechanoresponsive gene and potent regulator of naïve pluripotency.

They convincingly demonstrate that their approach via agarose microgels: Rex1::GFPd2 reporter cell line provides a quantitative real-time readout for naïve pluripotency. The microgel approach is nicely validated through live reporter quantification, as well as immunofluorescence stainings of cellular aggregates with other specific markers. Moreover, when encapsulated ESCs are subjected to chimera assays they colonize within the epiblast of the microinjected blastocyst with an impressive efficiency (more than 70%), further validating the use of the microgel platform. Experiments relating to the effects of volumetric confinement (vs 3D culture) are well controlled.

For the mechanistic insights, the authors combine fluorescence imaging techniques with chimera assays, loss and gain of function experiments via KO, OE approaches, and chemical perturbation. In addition, transcriptomic data is utilised to strengthen conclusions made. Statistics are solid.

I have several points for improving the validity of the conclusions:

- Can the authors comment on how the pressure exerted on ESCs in microgels relates to the pressure exerted by the extraembryonic tissues in the actual embryo? Can authors make a stiffness comparison against mammalian blastocyst?
- Can the authors comment on the "higher cytoplasmic β -catenin protein levels in 2D cultures on plastic" (page 8)?
- In figure 1, the loss of Rex1-GFP expression nicely demonstrates the naïve pluripotency exit, but could the authors also rule out random differentiation by confirming "metastable" pluripotency markers, e.g. Oct4 or Otx2-increased expression?
- Page 9: why 2D gel stiffness is being measured is not clear. Wouldn't it make sense to test gel stiffness in 3D? Please comment on it.
- While I am convinced of the role of plakoglobin in maintaining naïve state from the ESC-based experiments, I believe that the conclusions about the predicted function in in vivo development can be strengthened. For the authors' consideration, I suggest generating chimeric blastocysts using their β -catenin knockout in Plakoglobin-overexpressing ESCs and testing their pluripotent/differentiation potential by either in vitro culture or by in vivo embryo transfer and subsequent recovery of postimplantation stages.

Minor comment:

Page 11, T (Brachyury) should be introduced as a mesoderm marker, not a gastrulation marker.

Reviewer #2 (Remarks to the Author):

Synopsis:

In this manuscript by Kohler et al., the authors subject mouse embryonic stem cells (mESCs) to volumetric confinement by encapsulating the mESCs in agarose microgel droplets. They find that 3D colonies grown within the confines of encapsulating agarose upregulate expression of the β -catenin-like protein Plakoglobin, whereas 3D colonies grown in hanging drops of medium do not express detectable levels of Plakoglobin protein. A series of experiments are conducted to determine the role of Plakoglobin in mESC pluripotency. Overexpression of Plakoglobin, even in the absence of β -catenin, is capable of sustaining naive pluripotency in mESC maintained in serum + LIF (metastable maintenance conditions). Intriguingly, Plakoglobin expression is detected only in the epiblast cells of pre-implantation mouse and human blastocysts, and the authors suggest that Plakoglobin may act as a mechanoresponsive regulator of naive pluripotency in vivo.

Overview: 
The manuscript is well written, the experiments presented are well-controlled, and the data are convincing. Although more mechanistic insights regarding how Plakoglobin exerts its role at the transcriptional level would have strengthened the paper, I am satisfied that there is sufficient important new information provided, and further mechanistic studies are best left for subsequent investigations.

Overall, I have very few criticisms, as I found the study to be very well executed and articulated. My primary criticism is that more could be done to establish the universality of the role of Plakoglobin in naive pluripotency in mouse and human contexts.

Concerns:

1. A better description of the characterization of the CRISPR-mediated β -catenin knockout cell lines is required. The sequence of the knockout alleles and western blots confirming the absence of detectable β -catenin protein should be included.
2. More details regarding the culture of the agarose embedded cells is required. Are the agarose spheres kept in suspension in a bioreactor-like system with continual stirring, or are the embedded cells maintained in suspension in static conditions on low binding plates? This is important, as it has been shown that shear stress experienced by mESCs in bioreactors can influence the expression of pluripotency genes [Gareau, T. et al. Shear stress influences the pluripotency of murine embryonic stem cells in stirred suspension bioreactors. *J Tissue Eng Regen M* 8, 268–278 (2014)].
3. Although the authors have shown that epiblasts of preimplantation human blastocysts display the same pattern of Plakoglobin expression as observed in mouse blastocysts, they did not conduct experiments with human ESCs embedded in agarose to determine if they respond in the same way as mESCs with regard to an increase in Plakoglobin expression. Conducting similar experiments to those presented in Figure 2, particularly Figure 2e, would help support the hypothesis that Plakoglobin is a mechanosensitive regulator of pluripotency in the murine and human contexts. Also, it would be

informative to determine if overexpression of Plakoglobin in human pluripotent stem cells reinforces a state of naive pluripotency as observed with mESCs. Given improvements in culture conditions for maintaining human pluripotent stem cells in a naive state [e.g., Zimmerlin, L. et al. Tankyrase inhibition promotes a stable human naïve pluripotent state with improved functionality. *Development* 143, 4368–4380 (2016) and Khan, S. A. et al. Probing the signaling requirements for naive human pluripotency by high-throughput chemical screening. *Cell Reports* 35, 109233–109233 (2021)], the suggested experiments should be feasible.

Minor Concerns:

1. Throughout the manuscript, the authors describe the 2D cultures as being grown on plastic, which could suggest to readers that the cultures are plated directly on tissue culture plastic without any coating. I recommend that the authors refer to gelatin-coated plastic when first describing the methodology in the text (not just in the materials and methods) to ensure that readers understand that “on plastic” means “on gelatin-coated plastic”.
2. On page 4, “(GSK-3 β)” should be replaced by “(GSK-3)”.
3. On page 18, the figure legend for Extended Data Fig. 4c needs revision, as it does not reflect the data presented. That is, there is no quantification of fluorescent intensity along a white line presented in the data, rather, insets of magnified regions are shown.
4. On page 38, the abbreviation TCP is used without previously defining it. I assume it stands for tissue culture plastic, but it should be defined.

Reviewer #3 (Remarks to the Author):

In this manuscript, the authors investigated the effects of biomechanical environmental factors on mouse embryonic stem cells. More specifically the researchers focused on mimicking the volumetric compression that occurs in the blastocyst using agarose microgels to maintain naïve pluripotency. It was found that the confinement in microgels caused upregulation of Plakoglobin, along with an upregulation of many other naïve pluripotency factors. This was confirmed by comparison with scaffold-free hanging drop cell culture. Plakoglobin was found to be expressed in preimplantation mouse embryos. By forcing overexpression of Plakoglobin the researchers were able to maintain naïve pluripotency in the absence of B-catenin.

This manuscript builds upon previous knowledge that soft, non-degradable 3D matrices are beneficial to pluripotency by identifying Plakoglobin as an important mechanotransductive gene. The connection of volumetric confinement to embryogenesis is relevant to important biological phenomena.

NCOMMS-22-09994-T

Title: Plakoglobin is a mechanoresponsive regulator of naïve pluripotency

Authors: Timo N. Kohler, Joachim De Jonghe, Anna L. Ellerman¹, Ayaka Yanagida, Michael Herger, Erin M. Slatery, Katrin Fischer, Carla Mulas, Alex Winkel, Connor Ross, Sophie Bergmann, Kristian Franze, Kevin Chalut, Jennifer Nichols, Thorsten E. Boroviak, Florian Hollfelder

Major Concerns

Fig 1:

1. In panel b, what percentage of cell colonies in 2D showed protrusive morphology, and what percentage of cell colonies in 3D showed round dome-shaped morphology?
2. In panel c, immunofluorescence alone was not sufficient to prove the downregulation of OCT4 and KLF4 in encapsulated ESCs in differentiation media. It might be due to focusing on different planes of cell colonies. Other assays such as flow cytometry or Western Blotting are recommended.
3. In panel h, the chimeric embryos were stained 48 hrs after microinjection. Would a longer time of chimeric embryo culture lead to loss of naive pluripotency?

Fig S2:

4. In panel a, #3 Microgel was not correlated with other microgel samples. Also, one bulk microgel sample in panel b was not correlated with the others. This particular sample should not be considered a good replication.

Fig 3:

5. In panel c & d, the statistical analysis was only done for immunofluorescence intensity, without other quantitative methods. Moreover, only less than ten aggregates were tabulated, so the sample representation might not be ideal.
6. In panel c & d, on day 4, the difference between Microgel and HD was $p < 0.0001$ for β -catenin, whereas $p = 0.0007$ for plakoglobin. In the manuscript, the authors stated that ' β -catenin ... exhibited **slightly** elevated levels in microgels compared to hanging drops' (lines 175-177). This might be an understatement.

Fig 4.

7. Figure 4A. The meaning of MOR, ICM, EPI, PrE, and DIA are not explicitly stated prior to the mentioning in the paper, but later defined in Figure S4A caption. The definitions should be in Figure 4A as it appears prior.
8. Line 200 states Figure S4B is related to Ctnnb1 expression but this incorrect. Should be S4A.
9. Line 235 states a reciprocal relationship between Plakoglobin and β -catenin. Aside from the fluorescent data, is there any quantitative data provided? It is difficult to tell if that trend is present.

Fig S5:

10. In panel b, the authors only measured nuclear PG intensity. However, the previous staining suggested PG was expressing mainly at cell junctions. Would measuring the total PG expression show the same trend?
11. In the manuscript, the authors stated that 'all PG^{HIGH} clones displayed a uniformly positive Rex1-GFPd2 signal, indistinguishable from naïve ESCs cultured in 2i/LIF' (lines 242-243). However, the flow cytometry result for naïve ESCs cultured in 2i/LIF was not shown either in Fig 5E or S5A. Please add these results to both panels.

Fig 7:

12. In the manuscript, the authors stated that 'PG^{HIGH} ESCs cultured with LIF-only ... displayed homogenous (>99%) Rex1-GFPd2 levels, indistinguishable from 2i/LIF cells' (lines 314-315). However, the flow cytometry result for 2i/LIF cells was not shown in Fig 5b.
13. Line 323 says PH high could not maintain pluripotency and references Figure 7F. Figure 7F is shows data from PG low, it should be Figure 7E.
14. Figure 7J is meant to illustrate that PH high KO with PD remained pluripotent, however one of the clones did not survive past day 6. With only one clone remaining is it possible to say that it remained pluripotent? It would be nice to see multiple clones showing the tendency to remain pluripotent.

Minor Concerns:

1. In the RNA-seq (line 136) the type of culturing media is not explicitly stated. The prior section uses three different mediums making it unclear which medium is being used in this experiment.
2. Figure S3E. The plastic 2D culture has a much lower density of colonies. The coloring of the images make it unclear what is being expressed. Please add a color coated key for clarity.
3. Figure 3B. The microgel cultures are significantly smaller than the hanging drop culture. Later this is corrected for in Figure 3C but why not exclude potential aggregate size induced differences for this figure?
4. Figure S3A. Should this label be relative intensity, Plakoglobin/Hoechst versus Plakoglobin?
5. Figure 3C-D. Hoechst is label in black but expressed as purple in the figure. In Figure S3A the colors are indicated by the color of the labeling.
6. Figure 3D. The progression of 2+2 days is only shown at day 4. It would be interesting to see the day 3 as it will differ from the other conditions at day 3.
7. Acronym explanations should be added to the earliest appearance in panel a, instead of in Fig S4.
8. Figure 7A. Why the use of 2 PG high clones and only one low?
9. Line 770 'PO OE' should be PG OE.
10. Fig S1. Please add a scale bar in panel d.
11. Fig S2. In panel c, the graph title should be 'WNT/ β -catenin targets.' In panel e, please annotate what the green and blue colors represent respectively.
12. Delete blank lines (e.g., line 260).
13. The Discussion section can be divided into several paragraphs instead of one long paragraph.

Reviewer #4 (Remarks to the Author):

Summary: In this paper, Kohler and coauthors find that agarose microgel culture of mESCs support their naïve pluripotency even under serum-containing metastable conditions. By RNAseq comparing microgel vs. conventional culture conditions, they identify the adherens junction/desmosomal protein plakoglobin (Jup) as a differentially expressed gene that could explain the enhanced maintenance of pluripotency in this culture format. They further show that Plakoglobin expression is highest in these 3D systems at the pre-implantation stage, as opposed to the post-implantation blastocyst stage. Using gain- and loss-of-function experiments, the authors demonstrate that Jup overexpression is sufficient to induce naïve pluripotency under metastable conditions, even when beta-catenin is knocked out. Overall, I find these findings to be well supported by the rigorous experimentation in the manuscript, and recommend this paper for publication at Nature Communications with the following revisions:

Major Revisions

1. The authors make the claim that Jup is upregulated in microgel culture due to molecular crowding, and not stiffness. To support this claim, they present data from hanging microdrop culture and 2D soft PDMS substrates.^[1]
 - a. More experimental detail is required on the PDMS substrates – were these commercially purchased and were they coated with ECM prior to use? Important to the interpretation of these results is whether or not cells were adherent to the PDMS substrates.^[1]
 - b. To confirm that molecular confinement is indeed the cue, the authors should encapsulate mESC colonies growing on plastic in a bulk agarose gel to induce spatial confinement and switch to metastable conditions.
 - c. The authors measure a stiffness change in their agarose material with swelling, which is what would induce mechanical compression / crowding in the mESC colonies contributing to their stiffness. Can the swelling of the microgels be manipulated to show altered Jup expression as a direct result?
 - d. In our group's hands, commercially available PDMS substrates have not imaged well in the red- and far-red fluorescence channels. Some kind of positive control for Jup expression on PDMS would be welcome here to confirm that the results are not just due to selective wavelength absorbance. Alternatively, a western blot could be a more reliable means of quantitation for this and other assays.
2. For the studies showing that plakoglobin maintains naïve pluripotency independently to beta-catenin, the authors only conduct KO and inhibitor studies on beta-Catenin on the overexpressor PGHIGH clones and conclude that Plakoglobin is independent of beta-catenin signaling.
 - a. Is it plakoglobin itself or the overexpression of the protein that can maintain pluripotency independently to beta-catenin?
 - b. Would non-overexpressing cells in 3D culture with beta-catenin inhibition or KO yield the same results? If not, what is the missing piece?^[1]

Minor Revisions

1. Methods for testing statistical significance are largely absent from the paper. These should be included for all studies.
2. Figure 1B — If gels are spherical, why do cells create a dome-like structure? Are they attached to the plate?
3. Figure 2D-F – Certain conditions only have n=2. Are any of these results significant?
4. Figure 6 — For the scRNA results shown, were RGD2 cells in microgel culture under 2i/LIF culture also assayed? If not, why not?
5. Figure 6 – What are the microgel vs. TCP differentially expressed genes under metastable conditions, and do they align with the results from bulk RNAseq?^[1]
6. Due to the pink and blue shading representing different things between hanging drops and microgels in figure 3A, to a reviewer not familiar with this culture modality it appeared as though the hanging drops were some sort of cell aggregate in an agarose microwell, which is not the case. Slightly adjusting the shading or colors to specify what is media/gel/pbs would be useful here.
7. Rather than label every single DE gene in Figure 2C (which is challenging to read), providing a table and labeling only the bolded gene names would improve readability of this important data.

8. For the PCA plots in Figures S2A and 6C, labeling the PC axes with the percentage of variance explained is recommended as it is important to the interpretation of this kind of analysis.
9. The authors briefly touch on why volumetric confinement in the blastocyst state is important, yet it is confusing how volumetric confinement plays a role in the epiblast state, which is the state most of the studies are done on. Given that there is no fluid-cavity in these early stages, where is the "confinement" coming from? More discussion is warranted.
10. For the AFM measurements, the spring constant of the cantilever should be included. Also, 37mm bead should be 37 microns.

Responses for manuscript NCOMMS-22-09994-T

We thank the reviewers for their critical reading and their instructive comments that were very helpful for triggering additional experiments. We list the changes made in response to individual comments in the following pages and enclose an annotated copy of the main text for the reviewers, in which all textual changes are highlighted in red.

Reviewer #1:

Overall, this is an excellent study, the findings are highly interesting and well documented, and I support the publication.

The data is beautifully presented and the article is well written.

Despite a substantial body of work defining the chemical and cytokine-mediated signals that control pluripotency transcriptional programs, biomechanical cues and mechanoresponsive genes as emerging key players are ill-defined. The authors of this paper elegantly utilise an encapsulated microgel platform in which mESCs grow under self-renewing or differentiating conditions and display morphological development. Through this work, they identify Plakoglobin (a vertebrate homolog of b-catenin) as a mechanoresponsive gene and potent regulator of naïve pluripotency.

They convincingly demonstrate that their approach via agarose microgels: Rex1::GFPd2 reporter cell line provides a quantitative real-time readout for naïve pluripotency. The microgel approach is nicely validated through live reporter quantification, as well as immunofluorescence stainings of cellular aggregates with other specific markers. Moreover, when encapsulated ESCs are subjected to chimera assays they colonize within the epiblast of the microinjected blastocyst with an impressive efficiency (more than 70%), further validating the use of the microgel platform. Experiments relating to the effects of volumetric confinement (vs 3D culture) are well controlled.

For the mechanistic insights, the authors combine fluorescence imaging techniques with chimera assays, loss and gain of function experiments via KO, OE approaches, and chemical perturbation. In addition, transcriptomic data is utilised to strengthen conclusions made. Statistics are solid.

I have several points for improving the validity of the conclusions:

- Can the authors comment on how the pressure exerted on ESCs in microgels relates to the pressure exerted by the extraembryonic tissues in the actual embryo? Can authors make a stiffness comparison against mammalian blastocyst?

Our atomic force microscopy measurements of the agarose show that the microgels' stiffness is in the range of 400 Pa (Figure S1B). This value is in line with published measurements in the literature: the pressure in the mammalian blastocyst's cavity is around 500 Pa (without zona pellucida, Wang et al., 2018) (Wang et al. 2018) and 1400 to 1500 Pa (with zona pellucida) (Chii Jou Chan et al. 2019; Leonavicius et al. 2018). Hence, the microgels mimic the biophysical parameters more accurately than conventional tissue culture plastic - a hard 2D surface with stiffness in the GPa range (Kolahi et al. 2012). However, a detailed analysis of how pressure changes upon cell proliferation within the microgels will require further investigation outside the scope of this study.

- Can the authors comment on the “higher cytoplasmic b-catenin protein levels in 2D cultures on plastic” (page 8)?

In order to remove a possible misunderstanding the original sentence has been corrected to “ β -catenin was expressed in ESCs cultured in both conditions, although we noted reduced cortical protein levels in microgel cultured cells”. Our intention was to highlight the reduced β -catenin levels compared to the standard 2D control cells. Given that we are seeing an upregulation of Plakoglobin in microgel cultured cells the reduction of β -catenin is consistent with other studies showing a degradation of β -catenin upon Plakoglobin stabilisation (Salomon et al. 1997) and vice versa a compensatory upregulation of Plakoglobin upon β -catenin knockout (Lyashenko et al. 2011).

- In figure 1, the loss of Rex1-GFP expression nicely demonstrates the naïve pluripotency exit, but could the authors also rule out random differentiation by confirming “metastable” pluripotency markers, e.g. Oct4 or Otx2-increased expression?

We agree with the reviewer that the loss of the Rex1-GFPd2 reporter alone is not sufficient to confirm differentiation. In the original manuscript we confirmed downregulation of OCT4 and KLF4 upon culture in N2B27 (differentiation medium) via immunofluorescence analysis by confocal microscopy in Fig. 1c. We have added a protein analysis (whole lysate) by Western blot for OTX2, OCT4 and also KLF4 (as additionally requested by reviewer 3). These additional data were added to Fig. S1e.

Reproduction of Fig.S1e in the main body of the manuscript.

- Page 9: why 2D gel stiffness is being measured is not clear. Wouldn't it make sense to test gel stiffness in 3D? Please comment on it.

The 2D gels mentioned on page 9 refer to the commercially available PDMS gels (ibidi, #81291) that are offered in three different stiffnesses (1.5, 15.0 and 28.0 kPa). In this particular experiment we were investigating if changes in the stiffness of 2D surfaces would elicit the same effect as the microgel encapsulation. However, we did not observe any differences between the 2D gels and tissue culture plastic.

The stiffness of agarose used for the 3D microgels was measured via atomic force microscopy on bulk agarose gels. Unfortunately, we were limited to using 1.5% agarose because the physical properties make the agarose unsuitable for microfluidics at higher or lower concentrations.

- While I am convinced of the role of plakoglobin in maintaining naïve state from the ESC-based experiments, I believe that the conclusions about the predicted function in in vivo development can be strengthened. For the authors' consideration, I suggest generating chimeric blastocysts using their b-catenin knockout in Plakoglobin-overexpressing ESCs and testing their pluripotent/differentiation potential by either in vitro culture or by in vivo embryo transfer and subsequent recovery of postimplantation stages.

We have followed the suggestion of the reviewer and created chimeric blastocysts by aggregating Plakoglobin-overexpressing / β -catenin-knockout (PG^{HIGH} *Ctnnb1* KO#12) with uncompacted 8-cell stage embryos with subsequent *in vitro* culture until early post-implantation stages. We stained the chimeric embryos for Nanog (Pluripotency marker) and OTX2 (early post-implantation marker). Strikingly, we found that PG^{HIGH} *Ctnnb1* KO#12, in contrast to wild type cells, would co-express Nanog and OTX2. This suggests that Plakoglobin overexpression indeed stabilised the pluripotency gene regulatory network since Nanog and OTX2 are normally expressed in a mutually exclusive fashion (Acampora et al. 2016; Weberling and Zernicka-Goetz 2021). These new data were included in Fig. 7j and S7h&i.

Reproduction of Fig.7j in the main body of the manuscript.

Minor comment:

Page 11, T (*Brachyury*) should be introduced as a mesoderm marker, not a gastrulation marker.

We corrected the wording to “mesoderm” in the manuscript.

Reviewer #2:

Synopsis:

In this manuscript by Kohler et al., the authors subject mouse embryonic stem cells (mESCs) to volumetric confinement by encapsulating the mESCs in agarose microgel droplets. They find that 3D colonies grown within the confines of encapsulating agarose upregulate expression of the β -catenin-like protein Plakoglobin, whereas 3D colonies grown in hanging drops of medium do not express detectable levels of Plakoglobin protein. A series of experiments are conducted to determine the role of Plakoglobin in mESC pluripotency. Overexpression of Plakoglobin, even in the absence of β -catenin, is capable of sustaining naive pluripotency in mESC maintained in serum + LIF (metastable maintenance conditions). Intriguingly, Plakoglobin expression is detected only in the epiblast cells of pre-implantation mouse and human blastocysts, and the authors suggest that Plakoglobin may act as a mechanoresponsive regulator of naive pluripotency in vivo.

Overview:

The manuscript is well written, the experiments presented are well-controlled, and the data are convincing. Although more mechanistic insights regarding how Plakoglobin exerts its role at the transcriptional level would have strengthened the paper, I am satisfied that there is sufficient important new information provided, and further mechanistic studies are best left for subsequent investigations.

Overall, I have very few criticisms, as I found the study to be very well executed and articulated. My primary criticism is that more could be done to establish the universality of the role of Plakoglobin in naive pluripotency in mouse and human contexts.

Concerns:

1. A better description of the characterization of the CRISPR-mediated β -catenin knockout cell lines is required. The sequence of the knockout alleles and western blots confirming the absence of detectable β -catenin protein should be included.

The *Ctnnb1* knockout cell lines were generated with commercially available CRISPR/Cas9 *Ctnnb1* knockout plasmids (Santa Cruz, sc-419477, see methods), where *Ctnnb1* was targeted by a mixture of the following 3 guide RNAs (gRNAs):

A: ATGAGCAGCGTCAAAGTGGC

B: AGCTACTTGCTCTTGCGTGA

C: AAAATGGCAGTGCGCCTAGC

The gRNA sequences were added to the methods section.

Sanger sequencing confirmed the successful knockout of β -catenin in the clones #2 (B-del94 (OF), C-ins1 (OF)), #10 (B-del2 (OF), C-del38 (OF)) and #12 (B-ins76, B-del2 (OF)). The sequencing results were added as Fig. S8. Additionally, a Western blot was performed to confirm the absence of β -catenin protein in all the clones. These additional Western blot data were added to Fig. S7e. Long exposure (3 minutes) of the membrane revealed low amounts of a presumably truncated fragment of β -catenin in clone #2. In response, we added clone #12 and repeated the experiments.

Reproduction of Fig.S7e in the main body of the manuscript.

2. More details regarding the culture of the agarose embedded cells is required. Are the agarose spheres kept in suspension in a bioreactor-like system with continual stirring, or are the embedded cells maintained in suspension in static conditions on low binding plates? This is important, as it has been shown that shear stress experienced by mESCs in bioreactors can influence the expression of pluripotency genes [Gareau, T. et al. Shear stress influences the pluripotency of murine embryonic stem cells in stirred suspension bioreactors. *J Tissue Eng Regen M* 8, 268–278 (2014)].

We have integrated a more expanded description in the text “were cultured in a static suspension culture” (caption of Fig. 1) and added a method section for “Cell culture in microgels”. In contrast to Gareau et al., in our system embryonic stem cells are encapsulated in microgels and kept as a static suspension culture in conventional tissue culture plates. Hence, we do not expect to see the effects of shear stress on pluripotency as described by Gareau et al.

3. Although the authors have shown that epiblasts of preimplantation human blastocysts display the same pattern of Plakoglobin expression as observed in mouse blastocysts, they did not conduct experiments with human ESCs embedded in agarose to determine if they respond in the same way as mESCs with regard to an increase in Plakoglobin expression. Conducting similar experiments to those presented in Figure 2, particularly Figure 2e, would help support the hypothesis that Plakoglobin is a mechanosensitive regulator of pluripotency in the murine and human contexts. Also, it would be informative to determine if overexpression of Plakoglobin in human pluripotent stem cells reinforces a state of naive pluripotency as observed with mESCs. Given improvements in culture conditions for maintaining human pluripotent stem cells in a naive state [e.g., Zimmerlin, L. et al. Tankyrase inhibition promotes a stable human naive pluripotent state with improved functionality. *Development* 143, 4368–4380 (2016) and Khan, S. A. et al. Probing the signaling requirements for naive human pluripotency by high-throughput chemical screening. *Cell Reports* 35, 109233–109233 (2021)], the suggested experiments should be feasible.

We thank the reviewer for these excellent suggestions. As correctly mentioned, culture conditions for naive human pluripotent stem cells (hPSCs) have been greatly improved since first described by Gafni and colleagues (Gafni et al. 2013). Their naive human stem cell medium (NHSM) as well as the media “t2iLGö” (Takashima et al. 2015) and “5i/L/A” (Theunissen et al. 2014) described by other groups contain the small molecule CHIR99021; a GSK-3 inhibitor and WNT signalling agonist that stabilises β -catenin. These were based on the fact that β -catenin signalling is beneficial for the maintenance of naive mouse embryonic stem cells (Ying et al. 2008). However, since the beneficial effects of WNT signalling on the naive pluripotency network appear to be a mouse specific feature, researchers recently moved away from using the WNT agonist CHIR99021 and instead added the WNT antagonist XAV-939 (Huang et al. 2009) to the stabilise the culture of naive human pluripotent stem cells (Guo et al. 2017; Bredenkamp et al. 2019). Yet, others also report that Wnt/ β -catenin signalling promotes self-renewal and inhibits the primed state transition in naive human embryonic stem cells (Xu et al. 2016). Evidently, the role of WNT/ β -catenin signalling in the context of naive pluripotency in humans is still unclear.

We have previously encapsulated hPSCs (Schindler et al. 2021). However, in contrast to mouse embryonic stem cells encapsulated hPSCs (naive and primed) require initial Rho-associated protein kinase (ROCK) inhibition and the addition of Matrigel (Schindler et al. 2021; Watanabe et al. 2007). These chemically undefined conditions and different signalling requirements from mouse embryonic stem cells make them currently unsuited for direct comparison.

In light of these findings, the generation of Plakoglobin overexpressing human pluripotent stem cells will eventually be of great value to see if the beneficial effects on pluripotency are conserved across species.

Although we agree that both of the points raised are interesting, we believe that the suggested experiments are out of the scope of this study as establishment of improved culture conditions and clonal cell line derivations will take several months.

Instead we addressed the concern of the reviewer in the following experiments:

First, we have investigated Plakoglobin expression in mouse epiblast-derived stem cells (EpiSCs) which represent the post-implantation epiblast *in vitro*, and are known to be similar to conventional hPSCs (Brons et al. 2007; Tesar et al. 2007). Consistent with the downregulation of Plakoglobin in the post-implantation epiblast *in vivo*, EpiSCs do not express Plakoglobin. However, β -catenin and E-cadherin remained present.

Reproduction of Fig. S4c in the main body of the manuscript.

As we had observed an upregulation of Plakoglobin in the epiblast of human blastocysts we were also interested in investigating Plakoglobin expression upon implantation. However, early human post-implantation samples are inaccessible.

Reproduction of Fig. 4i in the main body of the manuscript.

Instead, non-human primate post-implantation marmoset embryos (Bergmann et al. 2022) were used as a comparative benchmark. We stained the post-implantation embryos for β -catenin and Plakoglobin and found their expression to be consistent with the *in vivo* and *in vitro* mouse data – Plakoglobin was downregulated during the transition to primed pluripotency whereas β -catenin remained expressed. Additionally, we analysed Plakoglobin levels via immunofluorescence in different human pluripotent stem cells (hPSCs) corresponding to the pre- and post-implantation stages, respectively. Naive derived hPSCs (Guo et al. 2016) displayed the highest levels of cortical Plakoglobin which was greatly reduced upon capacitation (Rostovskaya, Stirparo, and Smith 2019) and absent in primed hPSCs (Thomson et al. 1998).

Reproduction of Fig. 4j in the main body of the manuscript.

Together these additional data further strengthen the link between naive pluripotency and Plakoglobin in the mouse and human context. These new data were included in Fig. 4i,j, S4c,d,g-i.

Minor Concerns:

1. Throughout the manuscript, the authors describe the 2D cultures as being grown on plastic, which could suggest to readers that the cultures are plated directly on tissue culture plastic without any coating. I recommend that the authors refer to gelatin-coated plastic when first describing the methodology in the text (not just in the materials and methods) to ensure that readers understand that “on plastic” means “on gelatin-coated plastic”.

Yes, when conventionally cultured all cells were maintained on gelatine-coated tissue culture plastic. We have added “gelatin-coated plastic” on page 6 when introducing the methodology for the first time.

2. On page 4, “(GSK-3 β)” should be replaced by “(GSK-3)”.

The text has been adjusted to “GSK-3” in the manuscript.

3. On page 18, the figure legend for Extended Data Fig. 4c needs revision, as it does not reflect the data presented. That is, there is no quantification of fluorescent intensity along a white line presented in the data, rather, insets of magnified regions are shown.

We thank the reviewer for spotting this error. The figure caption has been corrected.

4. On page 38, the abbreviation TCP is used without previously defining it. I assume it stands for tissue culture plastic, but it should be defined.

We removed the abbreviation “TCP” altogether since we did not use it anywhere else in the manuscript. We also added on page 6, that tissue culture plastic was coated with gelatin.

Reviewer #3:

In this manuscript, the authors investigated the effects of biomechanical environmental factors on mouse embryonic stem cells. More specifically the researchers focused on mimicking the volumetric compression that occurs in the blastocyst using agarose microgels to maintain naïve pluripotency. It was found that the confinement in microgels caused upregulation of Plakoglobin, along with an upregulation of many other naïve pluripotency factors. This was confirmed by comparison with scaffold-free hanging drop cell culture. Plakoglobin was found to be expressed in preimplantation mouse embryos. By forcing overexpression of Plakoglobin the researchers were able to maintain naïve pluripotency in the absence of B-catenin.

This manuscript builds upon previous knowledge that soft, non-degradable 3D matrices are beneficial to pluripotency by identifying Plakoglobin as an important mechanotransductive gene. The connection of volumetric confinement to embryogenesis is relevant to important biological phenomena.

Major Concerns

Fig 1:

1. In panel b, what percentage of cell colonies in 2D showed protrusive morphology, and what percentage of cell colonies in 3D showed round dome-shaped morphology?

To address the question of the reviewer, we have added lower magnification overview images of mouse embryonic stem cells cultured on tissue culture plastic and after microgel encapsulation grown in 2i/LIF or serum/LIF. Only cells grown in serum/LIF on tissue culture plastic show the protrusive morphology. We have added a quantification of circularity (circularity was measured in ImageJ/FIJI where $\text{circularity} = 4\pi(\text{area}/\text{perimeter}^2)$). A value of 1.0 indicates a perfect circle). Based on an arbitrary circularity index cut-off of 0.75, we observe the following distribution (using the data in Fig. 1b):

Reproduction of Fig. 1b in the main body of the manuscript.

Plastic (serum/LIF) \approx 21.5%

Microgel (serum/LIF) \approx 96.4%

Plastic and Microgel (2i/LIF) = 100%

These new data were included in Fig. 1b.

2. In panel c, immunofluorescence alone was not sufficient to prove the downregulation of OCT4 and KLF4 in encapsulated ESCs in differentiation media. It might be due to focusing on different planes of cell colonies. Other assays such as flow cytometry or Western Blotting are recommended.

We have followed the reviewer's suggestions and performed a whole lysate western blot analysis of ESCs in 2i/LIF (pluripotent) and in N2B27 (differentiation) on tissue culture plastic and in microgels for the naive pluripotency marker KLF4 and the general pluripotency marker OCT4. Additionally, we added the peri-implantation/differentiation marker OTX2 (asked for by reviewer 1).

These new data were included in Fig. S1e.

Reproduction of Fig. S1e in the main body of the manuscript.

3. In panel h, the chimeric embryos were stained 48 hrs after microinjection. Would a longer time of chimeric embryo culture lead to loss of naive pluripotency?

As the reviewer correctly stated, embryos were fixed and stained at the blastocyst stage (48 hours after cell injection) because we initially wanted to confirm the cells' ability to still form blastocyst chimaeras after microgel culture. Given our current understanding we don't have any ground to believe that the cells would not be able to exit naive pluripotency.

This is due to the following reasons:

- 1) Microgel encapsulated ESCs downregulate the naive pluripotency marker KLF4 and upregulate the peri-implantation marker OTX2 upon culture in N2B27 differentiation medium as shown in Fig. 1c. S1e.
- 2) Additionally, in response to a question asked by Reviewer 1 we generated chimeric embryos with Plakoglobin overexpression + *Ctnnb1* (β -catenin) knockout (PG-OE/*Ctnnb1*-KO) cells and cultured them *in vitro* until early post-implantation. The chimeric postimplantation epiblasts show epithelialization and formation of a central lumen, both signs of the exit from naive pluripotency (Shahbazi et al. 2017). However, in contrast to wild type control embryos the PG-OE/*Ctnnb1*-KO epiblasts contain cells that are double positive for the naive marker Nanog and the peri-implantation marker OTX2 suggesting that constitutively forced overexpression of Plakoglobin can delay differentiation. These new data were added to Fig. 7.

Fig S2:

4. In panel a, #3 Microgel was not correlated with other microgel samples. Also, one bulk microgel sample in panel b was not correlated with the others. This particular sample should not be considered a good replication.

The principal component 1 in Fig. S2b accounts for 46.5% of the variance whereas principal component 2 accounts for 18% of the variance in the dataset (2.6-fold lower). Because PC1 clearly separates both bulk RNA-seq populations, triplicates correlate well and do not affect the main findings in Fig. 2. However, we do agree that some technical variability between samples exist, and therefore have changed the wording on page 7 (line 138) from

“clearly separated” to “separated” to reflect the sample separation along PC1 while nuancing its clear-cut appearance in the text. Additionally, we have added the PC percentages to Fig. S2b.

Fig 3:

5. In panel c & d, the statistical analysis was only done for immunofluorescence intensity, without other quantitative methods. Moreover, only less than ten aggregates were tabulated, so the sample representation might not be ideal.

Due to the inherently low throughput of the hanging drop culture method (that kept sample sizes small) we have updated Fig. 3 with new data generated from suspension culture. Suspension culture enables higher throughput analysis whilst offering similar biophysical parameters to the hanging drop method: 3D culture in the absence of spatial confinement. Furthermore, due to a discontinuation of the previously used antibody we repeated the experiment with a recombinant, knockout-verified, monoclonal anti-Plakoglobin antibody (Abcam: ab184919).

Reproduction of Fig. 3a, c&d in the main body of the manuscript.

Plakoglobin and β -catenin displayed similar expression patterns as observed previously. Upon microgel encapsulation, Plakoglobin was upregulated whereas the levels of β -catenin were reduced. These new data were included in Fig. 3.

Reproduction of Fig. 3e in the main body of the manuscript.

In addition to a comment by reviewer 4 we tested if spatial confinement by a bulk agarose gel on cells cultured on conventional tissue culture plastic would yield the same upregulation of Plakoglobin. Consistent with the microgel-mediated confinement, the bulk gel also induced Plakoglobin expression. These new data were included in Fig. 3.

Reproduction of Fig. 3f in the main body of the manuscript.

6. In panel c & d, on day 4, the difference between Microgel and HD was $p < 0.0001$ for β -catenin, whereas $p = 0.0007$ for plakoglobin. In the manuscript, the authors stated that ' β -catenin ... exhibited slightly elevated levels in microgels compared to hanging drops' (lines 175-177). This might be an understatement.

We have updated the manuscript to reflect the data presented in Fig. 3, including the new data generated by suspension culture.

Fig 4.

7. Figure 4A. The meaning of MOR, ICM, EPI, PrE, and DIA are not explicitly stated prior to the mentioning in the paper, but later defined in Figure S4A caption. The definitions should be in Figure 4A as it appears prior.

We thank the reviewer for the suggestion and have added the definitions to the updated figures.

8. Line 200 states Figure S4B is related to Ctnnb1 expression but this incorrect. Should be S4A.

We thank the reviewer for spotting this error and have changed the text accordingly.

9. Line 235 states a reciprocal relationship between Plakoglobin and β -catenin. Aside from the fluorescent data, is there any quantitative data provided? It is difficult to tell if that trend is present.

We thank the reviewer for the comment and in response performed Western blot analysis to support our findings originally seen by immunofluorescence stainings. RGd2, PG^{LOW} and PG^{HIGH} cells were cultured either in serum/LIF or 2i/LIF. Whole cell lysates were analysed for Plakoglobin and β -catenin levels. In serum/LIF cultured cells, the reciprocal relationship becomes clearly visible. With increased amounts of Plakoglobin, β -catenin protein levels decreased. In PG^{HIGH} cells, β -catenin was undetectable. Interestingly, when cultured in 2i/LIF β -catenin levels were unaffected regardless of the amount of Plakoglobin suggesting that Plakoglobin leads to rapid degradation of β -catenin on protein level but does not affect its transcription and translation. These new data were included in Fig. 5e.

Reproduction of Fig. 5e in the main body of the manuscript.

Additionally, there are several studies that describe the reciprocal relationship between Plakoglobin and β -catenin in different systems, including mouse embryonic stem cells and embryos (Haegel et al. 1995; Salomon et al. 1997; Lyashenko et al. 2011).

Fig S5:

10. In panel b, the authors only measured nuclear PG intensity. However, the previous staining suggested PG was expressing mainly at cell junctions. Would measuring the total PG expression show the same trend?

As explained in the comment above, we have performed Western blot analysis for Plakoglobin in RGd2, PG^{LOW} and PG^{HIGH} cells that are consistent with our previous observations made by immunofluorescent analysis.

11. In the manuscript, the authors stated that ‘all PG^{HIGH} clones displayed a uniformly positive Rex1-GFPd2 signal, indistinguishable from naïve ESCs cultured in 2i/LIF’ (lines 242-243). However, the flow cytometry result for naïve ESCs cultured in 2i/LIF was not shown either in Fig 5E or S5A. Please add these results to both panels.

We have added the flow cytometry data of RGd2 mouse ESCs grown under naïve conditions (2i/LIF positive control) to figures Fig. 5f and Fig. S5a.

Reproduction of Fig. 5f and S5a in the main body of the manuscript.

Fig 7:

12. In the manuscript, the authors stated that ‘PGHIGH ESCs cultured with LIF-only ... displayed homogenous (>99%) Rex1-GFPd2 levels, indistinguishable from 2i/LIF cells’ (lines 314-315). However, the flow cytometry result for 2i/LIF cells was not shown in Fig 5b.

We thank the reviewer for the suggestions and assume they were referring to Fig. 7b (please let us know if this is not the case). The 0 h time point displays the level of REX1-GFP positive cells in 2i/LIF before being transferred into single inhibitor/activator conditions (as illustrated in Fig. 7a) . For further clarification we added an average 2i/LIF base line (99.6% REX1-GFP+) for the RGd2 cells in all 3 graphs (Fig. 7 b-d) (here highlighted in red).

Reproduction of Fig. 7a-d in the main body of the manuscript.

13. Line 323 says PH high could not maintain pluripotency and references Figure 7F. Figure 7F is shows data from PG low, it should be Figure 7E.

We changed the text accordingly.

14. Figure 7J is meant to illustrate that PH high KO with PD remained pluripotent, however one of the clones did not survive past day 6. With only one clone remaining is it possible to say that it remained pluripotent? It would be nice to see multiple clones showing the tendency to remain pluripotent.

To address this concern, we validated the β -catenin knockouts further. Although all clones initially appeared to be complete knockouts, long exposure of the membrane (3 minutes) revealed a small amount of apparently truncated β -catenin in clone #2. Therefore, we added clone #10, a confirmed full β -catenin knockout, repeated the experiments and extended measurements to 30 days.

Reproduction of Fig. S7e in the main body of the manuscript.

All clones (#2, #10 and #12) maintained the dome-shaped morphology associated with naive pluripotency when supplemented with LIF or PD alone.

Reproduction of Fig. 7g in the main body of the manuscript.

We then improved the procedure for clone #12 (and the newly added #10) that previously had not survived in PD, as the cells detached from the plate. Coating the tissue culture dishes with laminin (generally more “sticky” than gelatin coating) for a minimum of 4 passages improved this situation dramatically and enabled us to keep the clone in culture; same applies to clone #10. Afterwards, the cells were able to be cultured on gelatin coating again. Differences in coating are indicated by the asterisks and described in the figure caption. Although clone #10 and #12 underwent and initial reduction of Rex1-GFPd2 reporter activity, the signal recovered to around 70% similar to clone #2

Reproduction of Fig. 7h in the main body of the manuscript.

We continued to assess the pluripotency state by immunofluorescence analysis of all RGd2, PG^{HIGH} and all PG^{HIGH} Ctnnb1 KO clones (#2, #10 and #12). We stained for the general pluripotency marker OCT4 and the naive pluripotency marker KLF4. We found Plakoglobin to fully support general pluripotency and in combination with LIF partially naive pluripotency. We updated the text accordingly and included these new data in Fig. 7 and S7.

Reproduction of Fig. S7g in the main body of the manuscript.

Reproduction of Fig. 7i in the main body of the manuscript.

Minor Concerns:

1. In the RNA-seq (line 136) the type of culturing media is not explicitly stated. The prior section uses three different mediums making it unclear which medium is being used in this experiment.

We thank the reviewer for their comment, we have added the media composition to the main text (2i+LIF).

2. Figure S3E. The plastic 2D culture has a much lower density of colonies. The coloring of the images make it unclear what is being expressed. Please add a color coated key for clarity.

We assume the reviewer refers to Fig. S2F (formerly S2E at first submission) and have added a color code for clarification.

3. Figure 3B. The microgel cultures are significantly smaller than the hanging drop culture. Later this is corrected for in Figure 3C but why not exclude potential aggregate size induced differences for this figure?

We have updated Fig. 3 with new data. We kept the original experiment with the large hanging drops because this is also used as one of the most common formats to prepare mESCs for embryoid body formation. We then used uncoated tissue culture plates to maintain mESCs in suspension culture in a similar fashion to hanging drops but compatible with a more high-throughput analysis and aggregate sizes corresponding to the microgel cultured cells.

4. Figure S3A. Should this label be relative intensity, Plakoglobin/Hoechst versus Plakoglobin?

We have updated the Y-axis label of Fig. S3a accordingly.

5. Figure 3C-D. Hoechst is label in black but expressed as purple in the figure. In Figure S3A the colors are indicated by the color of the labeling.

The original panels have been replaced by new data and a color code has been added.

6. Figure 3D. The progression of 2+2 days is only shown at day 4. It would be interesting to see the day 3 as it will differ from the other conditions at day 3.

We agree with the reviewer that a more detailed analysis of Plakoglobin and β -catenin protein dynamics could be of interest. However, here we focus on Plakoglobin's role in regulating pluripotency and the additional data point would not help to resolve this question.

7. Acronym explanations should be added to the earliest appearance in panel a, instead of in Fig S4.

In the new version of the manuscript the RNA-seq meta analysis panels were all shifted to Fig. S4a,b. We added the acronym explanations directly to the new figure caption. MOR = morula, ICM = inner cell mass, EPI = epiblast, PrE = primitive endoderm, DIA-EPI = diapaused epiblast.

8. Figure 7A. Why the use of 2 PG high clones and only one low?

As our work in the earlier parts of the manuscript (e.g. single cell sequencing in Fig. 6) had shown that the beneficial effects on the pluripotency network were only apparent when Plakoglobin was expressed at high levels, we decided to only use one PG^{LOW} clone in this experiment. This enabled us to keep the cell culturing workload manageable (4 cell lines in 4 different media = 16 conditions), whilst still running the single inhibitor/activator experiments in parallel.

9. Line 770 'PO OE' should be PG OE.

The typing error has been corrected in the manuscript.

10. Fig S1. Please add a scale bar in panel d.

We have added a scale bar to Fig. S1c (former panel d).

11. Fig S2. In panel c, the graph title should be 'WNT/ β -catenin targets.' In panel e, please annotate what the green and blue colors represent respectively.

The typing error has been corrected in Fig. S2 panel c and a color code has been added to panel f (former panel e).

12. Delete blank lines (e.g., line 260).

We deleted the blank lines.

13. The Discussion section can be divided into several paragraphs instead of one long paragraph.

We separated the discussion into 3 paragraphs instead of one long one.

Reviewer #4:

Summary: In this paper, Kohler and coauthors find that agarose microgel culture of mESCs support their naïve pluripotency even under serum-containing metastable conditions. By RNAseq comparing microgel vs. conventional culture conditions, they identify the adherens junction/desmosomal protein plakoglobin (Jup) as a differentially expressed gene that could explain the enhanced maintenance of pluripotency in this culture format. They further show that Plakoglobin expression is highest in these 3D systems at the pre-implantation stage, as opposed to the post-implantation blastocyst stage. Using gain- and loss-of-function experiments, the authors demonstrate that Jup overexpression is sufficient to induce naïve pluripotency under metastable conditions, even when beta-catenin is knocked out. Overall, I find these findings to be well supported by the rigorous experimentation in the manuscript, and recommend this paper for publication at Nature Communications with the following revisions:

Major Revisions

1. The authors make the claim that Jup is upregulated in microgel culture due to molecular crowding, and not stiffness. To support this claim, they present data from hanging microdrop culture and 2D soft PDMS substrates.

a. More experimental detail is required on the PDMS substrates – were these commercially purchased and were they coated with ECM prior to use? Important to the interpretation of these results is whether or not cells were adherent to the PDMS substrates.

For the experiments on 2D PDMS gels we used commercially available products from Ibidi (Ibidi μ -Dish 35 mm Elastically Supported Surface Cat.No:81199). These confocal microscopy compatible dishes consist of a 40 μ m thin PDMS layer on top of a 100 μ m glass cover and are available with a stiffness of 1.5 kPa, 15 kPa, or 28 kPa. The 2D PDMS gels were coated with 12 μ g/mL Laminin so that the cells could adhere to the substrate. We added a more detailed experimental description to the methods section.

b. To confirm that molecular confinement is indeed the cue, the authors should encapsulate mESC colonies growing on plastic in a bulk agarose gel to induce spatial confinement and switch to metastable conditions.

We thank the reviewer for this interesting experimental suggestion. To address this recommendation, we grew mESCs in small tissue culture wells and overlaid them with 1.5% low-melting agarose. To keep the experiment consistent with our data from the microgel culture, we kept the cells in 2i/LIF medium. In line with the results from microgel encapsulated cells, we observed an upregulation of Plakoglobin upon confinement with a bulk gel on top of the 2D tissue culture plastic. These additional data were added to Fig. 3f and the experimental details were explained in the methods section.

Reproduction of Fig. 3f in the main body of the manuscript.

c. The authors measure a stiffness change in their agarose material with swelling, which is what would induce mechanical compression / crowding in the mESC colonies contributing to their stiffness. Can the swelling of the microgels be manipulated to show altered Jup expression as a direct result?

As the reviewer correctly states, we measured different stiffnesses for agarose submerged in medium vs non-submerged agarose via atomic force microscopy. We cannot harness this swelling induced stiffening to change the microenvironment of the encapsulated cells since the microgels are kept in suspension culture from the start of the experiment. The stiffness of the hydrogel correlates linearly with the amount of agarose used. Due to the use of microfluidics we are limited in the range of agarose percentages that can be used. In the future, different hydrogels (e.g. PEG-based gels) could be used to modulate the stiffness to a greater extent than agarose allows in a microfluidic system.

d. In our group's hands, commercially available PDMS substrates have not imaged well in the red- and far-red fluorescence channels. Some kind of positive control for Jup expression on PDMS would be welcome here to confirm that the results are not just due to selective wavelength absorbance. Alternatively, a western blot could be a more reliable means of quantitation for this and other assays.

We have not encountered the imaging problems described by the reviewer and found the commercially available (Ibidi μ -Dish 35 mm Elastically Supported Surface Cat.No:81199) 2D PDMS gels optically active in all wavelength ranges. We confirmed the anti-rabbit-647 antibody by staining naive mESCs for β -catenin, as it is expected to be expressed regardless of substrate stiffness. No selective wavelength absorbance was detected compared to regular tissue culture plastic.

Reviewer figure: Naive mESCs on plastic and on 2D PDMS gels stained for β -catenin.

Furthermore, we repeated the experiment with a C-terminal and an N-terminal Plakoglobin binding antibody. Both antibodies did not reveal the strong upregulation of Plakoglobin as seen in spatially confined mESCs. These additional data were included in Fig. S3b.

Reproduction of Fig. S3b in the main body of the manuscript.

2. For the studies showing that plakoglobin maintains naïve pluripotency independently to beta-catenin, the authors only conduct KO and inhibitor studies on beta-Catenin on the overexpressor PGHIGH clones and conclude that Plakoglobin is independent of beta-catenin signaling.

a. Is it plakoglobin itself or the overexpression of the protein that can maintain pluripotency independently to beta-catenin?

In our current system the overexpression of Plakoglobin enables the maintenance of pluripotency. Microgel culture of mESCs and low levels of overexpression (PG^{low}) lead to the accumulation of Plakoglobin at the cortex but were not able to re-establish the naïve pluripotent gene regulatory network in metastable serum/LIF conditions as shown by, immunofluorescence stainings, flow cytometric analysis of the Rex1-GFPd2 reporter (Fig. 5) and single-cell RNA-seq (Fig. 6). In contrast, high levels of Plakoglobin overexpression (PG^{high}) re-establish the naïve pluripotency network; presumably through accumulation of nuclear Plakoglobin upon saturation of the cortex.

Consequently, it might be possible that endogenous levels of Plakoglobin could re-establish naïve pluripotency if localised to the nucleus. Overexpressed Plakoglobin initially locates to cell-cell junctions at the cortex which might explain a lack of nuclear translocation and the inability to sustain pluripotency until cortical binding sites are fully saturated.

b. Would non-overexpressing cells in 3D culture with beta-catenin inhibition or KO yield the same results? If not, what is the missing piece?

To address the reviewer's question we encapsulated RGd2 cells (no Plakoglobin overexpressing) in microgels and cultured them in N2B27 base medium with single supplementation of LIF, PD or CH and +/-XAV (XAV is a tankyrase inhibitor that stops nuclear β -catenin accumulation and signalling). Encapsulation of cells does not substitute for Plakoglobin overexpression and mESCs rapidly exit naïve pluripotency when cultured with only one supplement (+/- XAV). This might be due to insufficient amounts of Plakoglobin or incorrect localization of the protein.

These additional data were included in Fig S7c.

Reproduction of Fig. S7c in the main body of the manuscript.

Minor Revisions

1. Methods for testing statistical significance are largely absent from the paper. These should be included for all studies.

We included the statistical testing methods in the figure captions.

2. Figure 1B — If gels are spherical, why do cells create a dome-like structure? Are they attached to the plate?

The colonies are indeed spherical and not dome-shaped. However, our intention was to clarify that microgel-cultured cells, irrespective of the media, are more similar to cells cultured in 2i/LIF but not serum/LIF. To clarify this misunderstanding, the sentence has been corrected to: "When metastable pluripotent (serum/LIF) ESCs were cultured in microgels, the cells formed tightly packed colonies with indistinguishable cell boundaries and acquired

a spherical morphology, similar to that seen in dome-shaped naïve cells (2i/LIF) and unlike the serum/LIF cells cultured on plastic”

3. Figure 2D-F – Certain conditions only have n=2. Are any of these results significant?

In Fig.2d bulk RNA sequencing was performed in triplicate and differential gene expression and statistical significance were determined via DESeq2 (Love, Huber, and Anders 2014). To further confirm the upregulation of Plakoglobin upon encapsulation (Fig.2e), we repeated the experiments in 2 cell lines (RGd2 and wild type) with a different Plakoglobin antibody (anti-Plakoglobin, Abcam: ab184919, N-terminal binding, recombinant, knockout verified). We found the upregulation of Plakoglobin to be consistent across the two different cell lines with our previous results. These new data were included in Fig. S2e.

Reproduction of Fig. S2e in the main body of the manuscript.

4. Figure 6 — For the scRNA results shown, were RGd2 cells in microgel culture under 2i/LIF culture also assayed? If not, why not?

We thank the reviewer for their comment. RGd2 cells in microgel culture under 2i/LIF were assayed exclusively using bulk RNA-sequencing. The two main reasons to explain this choice are: 1) ESC cells cultured in 2i/LIF conditions are homogeneous in their transcriptome profile (Kolodziejczyk et al. 2015; Marks et al. 2012) and are therefore more suited for measurements using sensitive bulk RNA sequencing rather than more loss prone single-cell methods, this is not true for samples cultured in serum/LIF, as the cells display transcriptomic heterogeneity which would be blurred in bulk measurements. 2) we believe that the sample for cells cultured in 2i/LIF on tissue culture plastic is a more suitable integration anchor point as many transcriptomic datasets describing the continuum of cell states between primed and naive pluripotency exist (Kolodziejczyk et al. 2015; Marks et al. 2012) quantifying expression changes from cells cultured in 2i/LIF on tissue culture plastic. Because cells cultured in microgels under serum/LIF fall transcriptionally between naive and primed pluripotency and to contrast our findings to the existing body of literature, we feel that single-cell sequencing under tissue culture plastic in 2i/LIF is suitable anchor point to describe divergences from naive-like pluripotency under serum/LIF conditions (with and without microgel encapsulation).

5. Figure 6 – What are the microgel vs. TCP differentially expressed genes under metastable conditions, and do they align with the results from bulk RNAseq?

The comparison between microgel and TCP under metastable conditions was not assessed using bulk RNA-seq, due to the heterogeneous nature of the sample in this set of culture condition (Kolodziejczyk et al. 2015; Marks et al. 2012), which further suggest an exploration of cell heterogeneity based on scRNA-seq. However, following the reviewer’s question we computed the differentially expressed genes between metastable pluripotent (serum/LIF) mESCs in microgels and their counterpart on plastic.

Reviewer figure: Volcano plot showing the differential expression between cells cultured in microgel and TCP under metastable conditions. A Wilcoxon rank sum test was used to compute differential expression, a \log_2 fold change cut-off of $|\geq 0.5|$ and a Bonferroni adjusted p -value $< 10^{-3}$ were used as a cut-off for significance. Positive values for \log_2 fold changes indicate higher expression in TCP (plastic), negative values indicate higher expression in microgels.

Metastable pluripotent mESCs on plastic displayed characteristic upregulation of actin (*Actb*), ECM-related genes (*Sparc*, *Col4a1* and *Col4a2*) and serum-induced background differentiation associated genes (*Anxa2*, *Tmsb4x*) which were absent in microgel encapsulated mESCs.

Reproduction of Fig. 6g and S6g in the main body of the manuscript.

6. Due to the pink and blue shading representing different things between hanging drops and microgels in figure 3A, to a reviewer not familiar with this culture modality it appeared as though the hanging drops were some sort of cell aggregate in an agarose microwell, which is not the case. Slightly adjusting the shading or colors to specify what is media/gel/pbs would be useful here.

We have slightly adjusted the colors of the PBS and agarose and updated the design of the schematic. Furthermore, we added more experimental detail to the methods section.

Reproduction of Fig. 3a in the main body of the manuscript.

7. Rather than label every single DE gene in Figure 2C (which is challenging to read), providing a table and labeling only the bolded gene names would improve readability of this important data.

We have followed the suggestion of the reviewer and added a table with differentially expressed genes as a supplementary table (table 1). However, we also kept the labels in the figure to provide maximal information to researchers with different focus areas to potentially quickly identify their genes of interest.

8. For the PCA plots in Figures S2A and 6C, labeling the PC axes with the percentage of variance explained is recommended as it is important to the interpretation of this kind of analysis.

We thank the reviewer for their comment. It is indeed beneficial for data interpretation to add the variance ratios explained by the two first principal components, we therefore added these numbers in the Fig. S2A (PC1=46.5%, PC2=18.0%) and Fig. 6C (PC1=20.0%, PC2=7.4%).

9. The authors briefly touch on why volumetric confinement in the blastocyst state is important, yet it is confusing how volumetric confinement plays a role in the epiblast state, which is the state most of the studies are done on. Given that there is no fluid-cavity in these early stages, where is the “confinement” coming from? More discussion is warranted.

Naive pluripotency is established in the epiblast of the pre-implantation embryo and coincides with the upregulation of Plakoglobin (Boroviak et al. 2014; Ohsugi et al. 1996). At this stage, the epiblast is confined between the polar trophectoderm and the primitive endoderm. Simultaneously, the blastocoel cavity expands and the pressure within the embryo raises to a maximum of around 1.5 kPa (Dumortier et al. 2019; Chii Jou Chan et al. 2019). For further clarification about the origins of the pressure within the blastocyst we have added another two review references to the discussion (Chii J. Chan and Hiiragi 2020; Schliffka and Maître 2019).

10. For the AFM measurements, the spring constant of the cantilever should be included. Also, 37 mm bead should be 37 microns.

The spelling mistake was corrected in the manuscript and the spring constant of the cantilever was added to the methods section. A NanoWorld cantilever was used with a nominal spring constant value of 0.03. By looking back into the data we spotted a calculation error which we corrected and replotted the data. Additionally, another replicate was added. These new data were included in Fig. S1b.

Reproduction of Fig. S1b in the main body of the manuscript.

References

- Acampora, Dario, Daniela Omodei, Giuseppe Petrosino, Arcomaria Garofalo, Marco Savarese, Vincenzo Nigro, Luca Giovanni Di Giovannantonio, Vincenzo Mercadante, and Antonio Simeone. 2016. "Loss of the Otx2-Binding Site in the Nanog Promoter Affects the Integrity of Embryonic Stem Cell Subtypes and Specification of Inner Cell Mass-Derived Epiblast." *Cell Reports* 15 (12): 2651–64.
- Bergmann, Sophie, Christopher A. Penfold, Erin Slatery, Dylan Siriwardena, Charis Drummer, Stephen Clark, Stanley E. Strawbridge, et al. 2022. "Spatial Profiling of Early Primate Gastrulation in Utero." *Nature* 609 (7925): 136–43.
- Boroviak, Thorsten, Remco Loos, Paul Bertone, Austin Smith, and Jennifer Nichols. 2014. "The Ability of Inner-Cell-Mass Cells to Self-Renew as Embryonic Stem Cells Is Acquired Following Epiblast Specification." *Nature Cell Biology* 16 (6): 516–28.
- Bredenkamp, Nicholas, Jian Yang, James Clarke, Giuliano Giuseppe Stirparo, Ferdinand von Meyenn, Sabine Dietmann, Duncan Baker, et al. 2019. "Wnt Inhibition Facilitates RNA-Mediated Reprogramming of Human Somatic Cells to Naive Pluripotency." *Stem Cell Reports* 13 (6): 1083–98.
- Brons, I. Gabrielle M., Lucy E. Smithers, Matthew W. B. Trotter, Peter Rugg-Gunn, Bowen Sun, Susana M. Chuva de Sousa Lopes, Sarah K. Howlett, et al. 2007. "Derivation of Pluripotent Epiblast Stem Cells from Mammalian Embryos." *Nature* 448 (7150): 191–95.
- Chan, Chii J., and Takashi Hiragi. 2020. "Integration of Luminal Pressure and Signalling in Tissue Self-Organization." *Development* 147 (5). <https://doi.org/10.1242/dev.181297>.
- Chan, Chii Jou, Maria Costanzo, Teresa Ruiz-Herrero, Gregor Mönke, Ryan J. Petrie, Martin Bergert, Alba Diz-Muñoz, L. Mahadevan, and Takashi Hiragi. 2019. "Hydraulic Control of Mammalian Embryo Size and Cell Fate." *Nature* 571 (7763): 112–16.
- Dumortier, Julien G., Mathieu Le Verge-Serandour, Anna Francesca Tortorelli, Annette Mielke, Ludmilla de Plater, Hervé Turlier, and Jean-Léon Maître. 2019. "Hydraulic Fracturing and Active Coarsening Position the Lumen of the Mouse Blastocyst." *Science* 365 (6452): 465–68.
- Gafni, Ohad, Leehee Weinberger, Abed Alfatah Mansour, Yair S. Manor, Elad Chomsky, Dalit Ben-Yosef, Yael Kalma, et al. 2013. "Derivation of Novel Human Ground State Naive Pluripotent Stem Cells." *Nature* 504 (7479): 282–86.
- Guo, Ge, Ferdinand von Meyenn, Maria Rostovskaya, James Clarke, Sabine Dietmann, Duncan Baker, Anna Sahakyan, et al. 2017. "Epigenetic Resetting of Human Pluripotency." *Development* 144 (15): 2748–63.
- Guo, Ge, Ferdinand von Meyenn, Fatima Santos, Yaoyao Chen, Wolf Reik, Paul Bertone,

- Austin Smith, and Jennifer Nichols. 2016. "Naive Pluripotent Stem Cells Derived Directly from Isolated Cells of the Human Inner Cell Mass." *Stem Cell Reports* 6 (4): 437–46.
- Haegel, H., L. Larue, M. Ohsugi, L. Fedorov, K. Herrenknecht, and R. Kemler. 1995. "Lack of Beta-Catenin Affects Mouse Development at Gastrulation." *Development* 121 (11): 3529–37.
- Huang, Shih-Min A., Yuji M. Mishina, Shanming Liu, Atwood Cheung, Frank Stegmeier, Gregory A. Michaud, Olga Charlat, et al. 2009. "Tankyrase Inhibition Stabilizes Axin and Antagonizes Wnt Signalling." *Nature* 461 (7264): 614–20.
- Kolahi, Kevin S., Annemarie Donjacour, Xiaowei Liu, Wingka Lin, Rhodel K. Simbulan, Enrrico Bloise, Emin Maltepe, and Paolo Rinaudo. 2012. "Effect of Substrate Stiffness on Early Mouse Embryo Development." *PLoS One* 7 (7): e41717.
- Kolodziejczyk, Aleksandra A., Jong Kyoung Kim, Jason C. H. Tsang, Tomislav Ilicic, Johan Henriksson, Kedar N. Natarajan, Alex C. Tuck, et al. 2015. "Single Cell RNA-Sequencing of Pluripotent States Unlocks Modular Transcriptional Variation." *Cell Stem Cell* 17 (4): 471–85.
- Leonavicius, Karolis, Christophe Royer, Chris Preece, Benjamin Davies, John S. Biggins, and Shankar Srinivas. 2018. "Mechanics of Mouse Blastocyst Hatching Revealed by a Hydrogel-Based Microdeformation Assay." *Proceedings of the National Academy of Sciences of the United States of America* 115 (41): 10375–80.
- Love, Michael I., Wolfgang Huber, and Simon Anders. 2014. "Moderated Estimation of Fold Change and Dispersion for RNA-Seq Data with DESeq2." *Genome Biology* 15 (12): 550.
- Lyashenko, Natalia, Markus Winter, Domenico Migliorini, Travis Biechele, Randall T. Moon, and Christine Hartmann. 2011. "Differential Requirement for the Dual Functions of β -Catenin in Embryonic Stem Cell Self-Renewal and Germ Layer Formation." *Nature Cell Biology* 13 (7): 753–61.
- Marks, Hendrik, Tüzer Kalkan, Roberta Menafrá, Sergey Denissov, Kenneth Jones, Helmut Hofemeister, Jennifer Nichols, et al. 2012. "The Transcriptional and Epigenomic Foundations of Ground State Pluripotency." *Cell* 149 (3): 590–604.
- Ohsugi, M., S. Y. Hwang, S. Butz, B. B. Knowles, D. Solter, and R. Kemler. 1996. "Expression and Cell Membrane Localization of Catenins during Mouse Preimplantation Development." *Developmental Dynamics: An Official Publication of the American Association of Anatomists* 206 (4): 391–402.
- Rostovskaya, Maria, Giuliano G. Stirparo, and Austin Smith. 2019. "Capacitation of Human Naïve Pluripotent Stem Cells for Multi-Lineage Differentiation." *Development* 146 (7). <https://doi.org/10.1242/dev.172916>.
- Salomon, Daniela, Paula A. Sacco, Sujata Guha Roy, Inbal Simcha, Keith R. Johnson, Margaret J. Wheelock, and Avri Ben-Ze'ev. 1997. "Regulation of β -Catenin Levels and Localization by Overexpression of Plakoglobin and Inhibition of the Ubiquitin-Proteasome System." *The Journal of Cell Biology* 139 (5): 1325–35.
- Schindler, Magdalena, Dylan Siriwardena, Timo N. Kohler, Anna L. Ellermann, Erin Slatery, Clara Munger, Florian Hollfelder, and Thorsten E. Boroviak. 2021. "Agarose Microgel Culture Delineates Lumenogenesis in Naive and Primed Human Pluripotent Stem Cells." *Stem Cell Reports* 16 (5): 1347–62.
- Schliffka, Markus Frederik, and Jean-Léon Maître. 2019. "Stay Hydrated: Basolateral Fluids Shaping Tissues." *Current Opinion in Genetics & Development* 57 (August): 70–77.
- Shahbazi, Marta N., Antonio Scialdone, Natalia Skorupska, Antonia Weberling, Gaëlle Recher, Meng Zhu, Agnieszka Jedrusik, et al. 2017. "Pluripotent State Transitions Coordinate Morphogenesis in Mouse and Human Embryos." *Nature* 552 (7684): 239–43.
- Takashima, Yasuhiro, Ge Guo, Remco Loos, Jennifer Nichols, Gabriella Ficz, Felix Krueger, David Oxley, et al. 2015. "Resetting Transcription Factor Control Circuitry toward Ground-State Pluripotency in Human." *Cell* 162 (2): 452–53.
- Tesar, Paul J., Josh G. Chenoweth, Frances A. Brook, Timothy J. Davies, Edward P. Evans, David L. Mack, Richard L. Gardner, and Ronald D. G. McKay. 2007. "New Cell Lines

- from Mouse Epiblast Share Defining Features with Human Embryonic Stem Cells.” *Nature* 448 (7150): 196–99.
- Theunissen, Thorold W., Benjamin E. Powell, Haoyi Wang, Maya Mitalipova, Dina A. Faddah, Jessica Reddy, Zi Peng Fan, et al. 2014. “Systematic Identification of Culture Conditions for Induction and Maintenance of Naive Human Pluripotency.” *Cell Stem Cell* 15 (4): 524–26.
- Thomson, J. A., J. Itskovitz-Eldor, S. S. Shapiro, M. A. Waknitz, J. J. Swiergiel, V. S. Marshall, and J. M. Jones. 1998. “Embryonic Stem Cell Lines Derived from Human Blastocysts.” *Science* 282 (5391): 1145–47.
- Wang, Xian, Zhuoran Zhang, Hirotaoka Tao, Jun Liu, Sevan Hopyan, and Yu Sun. 2018. “Characterizing Inner Pressure and Stiffness of Trophoblast and Inner Cell Mass of Blastocysts.” *Biophysical Journal* 115 (12): 2443–50.
- Weberling, Antonia, and Magdalena Zernicka-Goetz. 2021. “Trophectoderm Mechanics Direct Epiblast Shape upon Embryo Implantation.” *Cell Reports* 34 (3): 108655.
- Xu, Zhuojin, Aaron M. Robitaille, Jason D. Berndt, Kathryn C. Davidson, Karin A. Fischer, Julie Mathieu, Jennifer C. Potter, Hannele Ruohola-Baker, and Randall T. Moon. 2016. “Wnt/ β -Catenin Signaling Promotes Self-Renewal and Inhibits the Primed State Transition in Naïve Human Embryonic Stem Cells.” *Proceedings of the National Academy of Sciences* 113 (42): E6382–90.
- Ying, Qi-Long, Jason Wray, Jennifer Nichols, Laura Batlle-Morera, Bradley Doble, James Woodgett, Philip Cohen, and Austin Smith. 2008. “The Ground State of Embryonic Stem Cell Self-Renewal.” *Nature* 453 (7194): 519–23.

REVIEWERS' COMMENTS

Reviewer #1 (Remarks to the Author):

I appreciate the authors' efforts in revising their manuscript. All comments raised in my initial review have been carefully addressed by the authors. The revised manuscript shows substantial improvements with additional experimental data and textual clarifications that increase the overall readability and help to improve how convincing the arguments are. Overall my enthusiasm continues for this manuscript, and I believe the new information provided in this work is sufficient and important to the field.

Reviewer #2 (Remarks to the Author):

The authors have very thoroughly addressed my concerns, and I fully support publication of their manuscript.

Reviewer #3 (Remarks to the Author):

In this manuscript, the authors investigated the effects of biomechanical environmental factors on mouse embryonic stem cells. More specifically, the researchers focused on mimicking the volumetric compression in the blastocyte using agarose microgels to maintain naïve pluripotency. It was found that the confinement in microgels caused the upregulation of Plakoglobin, along with an upregulation of many other naïve pluripotency factors. This was confirmed by comparison with scaffold-free hanging drop cell culture. Plakoglobin was found to be expressed in preimplantation mouse embryos. By forcing overexpression of Plakoglobin, the researchers could maintain naïve pluripotency without B-catenin. This manuscript builds upon previous knowledge that soft, non-degradable 3D matrices benefit pluripotency by identifying Plakoglobin as an important mechanotransductive gene. The minor concerns from the first submission were adequately addressed, and the methodology is sound, providing new insight into stem cells and mechanobiology.

Reviewer #4 (Remarks to the Author):

The authors have satisfactorily addressed each of my prior concerns. I am supportive of publication at Nature Communications after the following *very* minor corrections are made:

Figure 1c/g, S3a, 5g/h/i – y axis labels are misspelled 'intesity'

S3b "Avergae"

2ab vs 2c – normalized vs normalised